# GPT-IMAGE-EDIT-1.5M: A MILLION-SCALE EDITING DATASET AND WHAT ACTUALLY WORKS TO TRAIN STRONG EDITORS

## ABSTRACT

Recent advancements in proprietary multimodal models such as GPT-Image-1 have set new standards for high fidelity, instruction guided image editing. However, their closed-source nature restricts open research and reproducibility. To bridge this gap, we introduce GPT-IMAGE-EDIT-1.5M, a publicly available dataset comprising over 1.5 million high-quality editing triplets systematically unified from OmniEdit, HQEdit, and UltraEdit. Our data curation pipeline leverages output regeneration and instruction rewriting to significantly enhance instruction following (IF) and perceptual quality (PQ), while relying only on simple geometric and instruction-level filters. We benchmark three MMDiT diffusion architectures—SD3 InstructPix2Pix (channel-wise conditioning), Flux with SigLIP (token-wise conditioning), and FluxKontext (token-wise conditioning) to analyze their robustness against IP degradation. Our results indicate that token-wise conditioning methods consistently outperform channel-wise conditioning. To ensure evaluation transparency, we specify when results involve thinking-rewritten prompts to avoid potential ambiguity. Moreover, we examine text encoders within a common frozen-encoder scenario, demonstrating that T5 embeddings consistently meet or exceed multimodal large language model (MLLM) embeddings, particularly with lengthier prompts. Simple linear or query-based integration methods, however, offer limited improvements, indicating deeper cross-modal fusion methods may be necessary. Fine-tuning FluxKontext on GPT-IMAGE-EDIT-1.5M achieves open-source performance competitive with GPT-Image-1 (**7.66**@GEdit-EN and **3.90**@ImgEdit-Full, with thinking-rewritten prompts; **8.97**@Complex-Edit). Our findings highlight critical interactions among instruction complexity, semantic alignment, and identity preservation, informing future directions in open-source image editing.

## 1 INTRODUCTION

Instruction-guided image editing is a fundamental task for generative AI, spurring significant progress in diffusion-based models such as InstructPix2Pix (Brooks et al., 2023), Prompt-to-Prompt (Hertz et al., 2022), SDEdit (Meng et al., 2021), and Imagic (Kawar et al., 2023). Proprietary models, notably GPT-Image-1 (Hurst et al., 2024), currently set the highest standards in instruction-following (IF) and perceptual quality (PQ). However, their closed-source nature severely restricts open research and reproducibility, creating a persistent gap between proprietary and open-source methods (Shi et al., 2024; Wang et al., 2025b).

A critical obstacle for open-source methods is the lack of large-scale, diverse, and well-aligned datasets. Existing datasets such as OmniEdit (Wei et al., 2025a), HQEdit (Hui et al., 2025), and UltraEdit (Zhao et al., 2024) frequently provide overly simplistic instructions or suffer from weak alignment between instructions and images. Consequently, open-source models trained on these datasets typically fail to achieve performance comparable to proprietary solutions.

To overcome these limitations, we introduce GPT-IMAGE-EDIT-1.5M, a unified dataset comprising over 1.5 million high-quality editing triplets (Fig. 1). Our streamlined pipeline leverages GPT-Image-1 to significantly enhance IF and PQ through *output regeneration* and *instruction rewriting*.

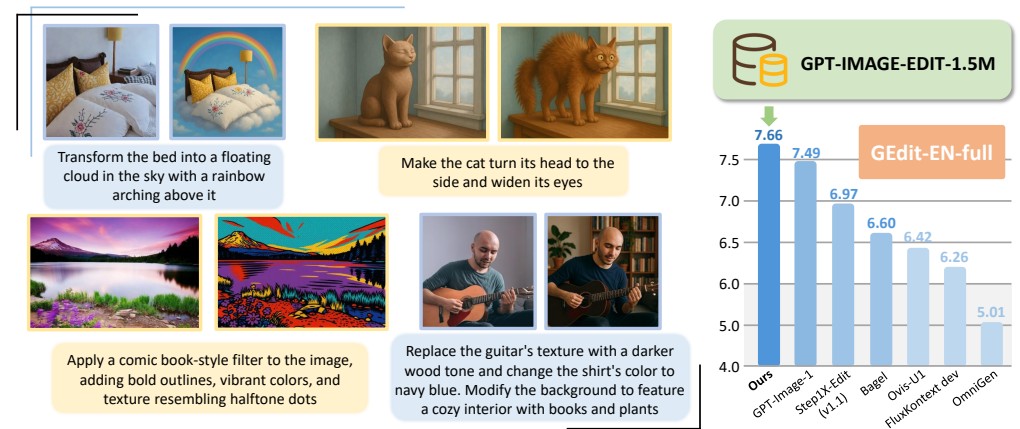

Figure 1: An overview of the GPT-IMAGE-EDIT-1.5M dataset. The figure presents qualitative examples showcasing diverse and complex instruction-guided edits. The bar chart demonstrates that a model fine-tuned on GPT-IMAGE-EDIT-1.5M achieves 7.66 on the GEdit-EN-full benchmark, surpassing existing open-source methods.

Unlike pipelines that aggressively curate away difficult identity-preservation (IP) cases, we apply only simple geometric checks and discard obvious catastrophic failures, without any IP- or text-specific pruning. As a result, GPT-IMAGE-EDIT-1.5M follows the natural IP and text behavior of GPT-generated edits and still contains realistic identity and text-rendering challenges. Our experiments show that even under this lightweight filtering, fine-tuning a 12B open editor on GPT-IMAGE-EDIT-1.5M is enough to approach or surpass GPT-Image-1 on several benchmarks, suggesting that more sophisticated IP-aware filters are a promising but optional next step.

Considering the inherent IP challenges, we systematically evaluate three diffusion architectures built upon MMDiT (Esser et al., 2024): SD3 InstructPix2Pix (Zhao et al., 2024) with channel-wise conditioning, and Flux with SigLIP (Lin et al., 2025) and FluxKontext (Labs et al., 2025), both employing token-wise conditioning. Our analyses consistently indicate that token-wise conditioning notably surpasses channel-wise methods across all evaluated metrics. This finding supports the idea that finer-grained token-level conditioning can more effectively manage semantic nuances and spatial alignment, essential for accurate instruction-guided edits.

Additionally, we examine text encoder strategies under a common practical constraint: frozen encoder parameters during fine-tuning. We observe that robust text-only encoders, such as T5, consistently match or exceed multimodal large language model (MLLM) embeddings, particularly with detailed, lengthy prompts. Furthermore, shallow integration methods like linear projections (Lin et al., 2025; Liu et al., 2025) or query-based connectors (Pan et al., 2025; Wei et al., 2025b) offer limited improvements, underscoring the need for deeper and more sophisticated cross-modal fusion methods (Tang et al., 2025; Deng et al., 2025; Xie et al., 2025).

Fine-tuning FluxKontext on GPT-IMAGE-EDIT-1.5M achieves open-source performance approaching proprietary GPT-Image-1, particularly when employing **7.66**@GEdit-EN, **3.90**@ImgEdit-Full with thinking-rewritten prompts, **8.97**@Complex-Edit. Rather than merely restating numerical results, our study provides nuanced insights into the relationships between instruction complexity, semantic alignment, and identity preservation, guiding future open-source advancements.

**Contribution**

- **Data:** We leverage GPT-Image-1 to build GPT-IMAGE-EDIT-1.5M, a unified dataset of over 1.5 million high-quality editing triplets, significantly enriching instruction diversity and alignment.

- **Conditioning Mechanism:** A systematic evaluation demonstrating the superiority of token-wise conditioning over channel-wise approaches for accurate and context-aware editing.

- **Text Encoder:** A detailed comparative analysis of text encoders under frozen encoder conditions, confirming the effectiveness of T5 embeddings and highlighting limitations in shallow MLLM integration methods.

• **Evaluation:** Comprehensive empirical evaluation across major benchmarks, clearly identifying strengths, weaknesses, and critical trade-offs necessary to advance open-source image editing.

## 2 RELATED WORKS

**Instruction-Guided Image Editing.** The task of instruction-guided image editing was established by pioneering works such as InstructPix2Pix (Brooks et al., 2023), Prompt-to-Prompt (Hertz et al., 2022), SDEdit (Meng et al., 2021), and Imagic (Kawar et al., 2023). InstructPix2Pix introduced a scalable, two-step approach: first leveraging GPT-3 (Brown et al., 2020) to generate synthetic instruction-image triplets, then using a diffusion model guided by Prompt-to-Prompt control (Hertz et al., 2022) to produce the corresponding image edits. Despite its foundational impact, the performance of these early models was constrained by the underlying diffusion architectures (U-Net-based latent diffusion models trained with CLIP (Radford et al., 2021)), limiting their photorealism and semantic precision (Rombach et al., 2022). This motivated subsequent research to pursue improvements in both dataset quality and architectural capability.

**Data-Centric Advancements.** Recognizing the critical role of data quality, recent approaches have prioritized sophisticated dataset curation. For instance, HQEdit (Hui et al., 2025) utilizes powerful proprietary models such as GPT-4V (Hurst et al., 2024) and DALL-E 3 (OpenAI, 2023) to generate more aligned and high-quality editing pairs. Concurrently, ShareGPT-4o-Image (Chen et al., 2025) demonstrates effective direct distillation from proprietary models, creating high-quality datasets explicitly designed to transfer advanced editing capabilities to smaller, open-source models. Aligning with this strategy, our work systematically leverages GPT-Image-1 to refine and unify large-scale datasets, significantly enhancing data alignment and diversity without complex design.

**Architectural Evolution: Diffusion and Flow Matching.** Generative model architectures have evolved considerably, transitioning from U-Net-based diffusion models (Rombach et al., 2022) to more scalable Transformer-based Diffusion Transformers (DiT) (Peebles & Xie, 2023). More recently, flow matching (FM) models (Lipman et al., 2022) have emerged as efficient alternatives, predicting continuous velocity fields to directly model complex distributions. Specifically, FLUX.1 Kontext (Labs et al., 2025) exemplifies a state-of-the-art FM-based architecture, efficiently unifying generation and editing through token-wise conditioning, demonstrating robust semantic and perceptual capabilities. We leverage this architecture due to its proven effectiveness and efficiency, particularly suited to instruction-guided editing tasks.

**Semantic Enhancement via Token-Wise Conditioning.** An essential improvement in multi-modal generative models has been the advancement in conditioning strategies—particularly token-wise versus channel-wise integration. State-of-the-art open-source models such as Step1X-Edit (Liu et al., 2025) and UniWorld-V1 (Lin et al., 2025) leverage token-wise conditioning schemes: Step1X-Edit utilizes Kontext-based token fusion, while UniWorld-V1 employs SigLIP-based token-wise integration, each conditioned on powerful multimodal large language models (MLLMs) like Qwen-VL (Bai et al., 2025) or LLaVA (Liu et al., 2023). These approaches significantly enhance semantic alignment and editing precision compared to earlier channel-wise methods. Our systematic exploration of these paradigms demonstrates clear advantages of token-wise conditioning in robustness and semantic fidelity, especially under realistic identity preservation (IP) challenges.

**Evaluation Benchmarks.** We evaluate on comprehensive benchmarks capturing diverse editing scenarios: *GEdit-Bench-EN (Full)* covers 11 distinct editing tasks with MLLM-based scoring (Liu et al., 2025); *ImgEdit (Full)* assesses across 9 task families using a unified pipeline (Ye et al., 2025); *Complex-Edit* evaluates compositional reasoning through chained edits (Yang et al., 2025); These benchmarks ensure rigorous evaluation across multiple dimensions (IF, IP, PQ), guiding the reliable assessment and comparison of editing model architectures.

## 3 DATA & METHOD

Our primary goal is to construct a large-scale, high-quality dataset to facilitate robust open-source instruction-guided image editing. To this end, we introduce a unified, minimalist pipeline for data

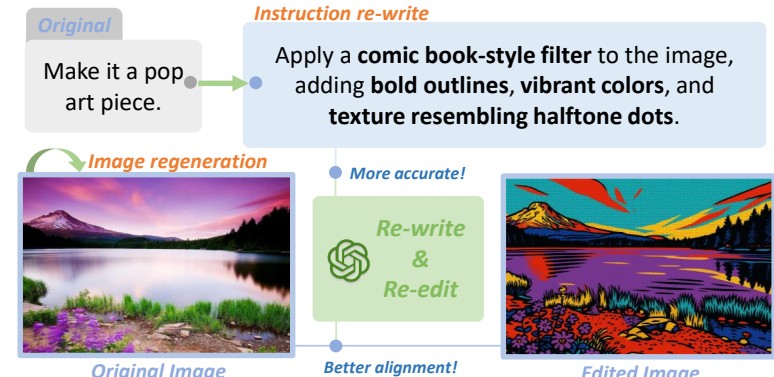

Figure 2: An overview of GPT-IMAGE-EDIT-1.5M data curation pipeline. We applied multiple methods to collect high-quality image-editing data. we re-write 10% of OmniEdit instructions to better match regenerated images, and the input images originally generated by DALL-E in HQEdit were re-synthesized by GPT-Image-1 for higher alignment.

curation as shown in Fig. 2, designed to produce well-aligned instruction–image pairs while leaving the identity-preservation (IP) behavior of GPT-Image-1 unchanged, thereby retaining the challenging IP cases typical of GPT-generated content. Given the IP challenges inherent to our dataset, we further investigate how different conditioning mechanisms and text encoder choices within MMDiT architectures influence editing quality and robustness. Below, we first describe our dataset curation process in detail, followed by an exploration of these key architectural decisions.

### 3.1 UNIFIED DATA CURATION AND EVALUATION PIPELINE

Our dataset curation process strategically integrates multiple methods to enhance the alignment, complexity, and quality of instruction-guided image editing data. We employ GPT-Image-1 to re-generate output images from existing instruction-image pairs, substantially improving visual fidelity and instruction-following accuracy. To address potential semantic drift from regeneration, we further utilize GPT-4o to selectively rewrite approximately 10% of OmniEdit instructions, ensuring that textual prompts precisely reflect the visual modifications in regenerated images.

Additionally, we introduce composite editing instructions of moderate complexity (three-step atomic edits, C3-level) (Yang et al., 2025) to approximately half of the OmniEdit dataset, enriching the dataset's realism and complexity. To upgrade the input quality of the HQ-Edit dataset, originally synthesized by DALL-E 3, we regenerate all inputs using GPT-Image-1, thereby enhancing visual quality and ensuring stronger alignment between instructions and images.

To maintain dataset consistency across varying aspect ratios, we implement a robust pad-and-crop procedure that standardizes images to three fixed ratios (1:1, 3:2, 2:3) without distortion. Post-generation, images are precisely cropped, with automated filtering mechanisms eliminating images containing artifacts or residual padding. Beyond these geometric checks, we do not rank or filter samples using face recognition, OCR, or other identity proxies; GPT-IMAGE-EDIT-1.5M therefore inherits the imperfect identity and scene-text behavior of GPT-Image-1, which we view as part of the target distribution rather than noise. Further details are presented in the Appendix B.

During evaluation, recognizing ambiguity in conventional short-text instructions, we employ GPT-5 at inference time to systematically rewrite raw benchmark prompts into clearly structured instructions. This rewriting step clarifies the intended edits by explicitly defining input conditions, desired edits, and expected outputs, while preserving the original images and evaluation scoring systems, thereby maintaining transparency and evaluation integrity. Comprehensive procedural details and examples are included in the Appendix D.

### 3.2 CONDITIONING PARADIGMS: CHANNEL-WISE VS. TOKEN-WISE

An MMDiT-based image editing model typically conditions the generation process on both image and text inputs. We explore three distinct conditioning paradigms within the broader MMDiT

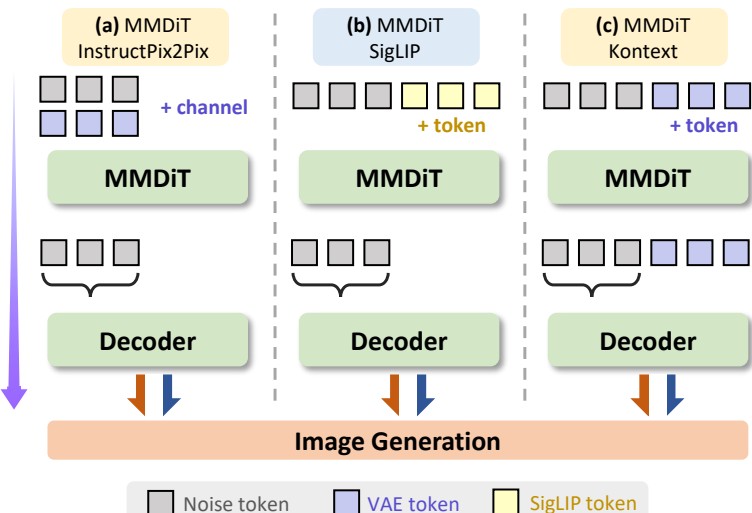

Figure 3: Conditioning paradigms comparison within the MMDiT architecture. (a) Channel-wise conditioning as in SD3 InstructPix2Pix concatenates conditioning information along input channels, subsequently compressing dimensionality within the model. (b) Flux with SigLIP employs token-wise merging via SigLIP visual features into textual embedding space, maintaining simplicity and strong semantic alignment. (c) FluxKontext leverages a robust dual-stream token-wise conditioning method, embedding visual and noise prediction tokens separately for enhanced precision and identity preservation, albeit at higher computational cost.

architecture family (Fig. 3), each varying by conditioning granularity, token fusion strategy, and computational complexity:

**SD3 InstructPix2Pix (Channel-wise Conditioning).** This method concatenates conditioning information directly along the channel dimension at input embedding layers, effectively increasing the embedding dimensionality. Post-processing within MMDiT embedding layers subsequently compresses these concatenated channels back to the original token dimension. While straightforward, this approach may suffer from redundancy and higher complexity in handling channel-wise concatenation, potentially limiting its robustness to spatial misalignments and minor semantic discrepancies.

**Flux with SigLIP (Token-wise Conditioning).** In the Flux architecture, extracted SigLIP visual tokens are first projected into the textual embedding space, then merged token-wise with text embeddings via Flux's distinctive hybrid dual-to-single stream strategy. Unlike traditional dual-stream approaches, Flux merges visual and textual tokens within its single-stream stage. Consequently, only noise prediction tokens are active in the final decoding layers, significantly simplifying the conditioning mechanism and promoting more robust semantic alignment and identity preservation.

**FluxKontext (Token-wise Conditioning).** FluxKontext adopts a comprehensive dual-stream conditioning framework, embedding noise prediction tokens alongside latent visual tokens (derived from a VAE) as separate image branches. These branches remain distinct yet are processed in parallel by MMDiT layers, effectively doubling computational demands compared to single-stream approaches. Despite increased complexity, FluxKontext consistently achieves strong performance across multiple open-source benchmarks, reflecting its precise and robust conditioning capabilities.

### 3.3 TEXT ENCODERS AND FUSION STRATEGIES

**Frozen-encoder setting.** In all our experiments, we keep the text encoders (T5 (Raffel et al., 2020) and Qwen2.5-VL-7B Bai et al. (2025)) frozen. Both encoder-decoder (T5) and decoder-only (MLLM) models are used purely as *encoders*, performing a single forward pass without autoregressive decoding. We only fine-tune the lightweight projection and fusion layers, ensuring fair comparison.

Let $E_{t5} \in \mathbb{R}^{L_{t5} \times d}$ denote embeddings from T5, and $E_{mllm} \in \mathbb{R}^{L_m \times d_m}$ from Qwen2.5-VL-7B. We use linear layer $W \in \mathbb{R}^{d_m \times d}$ to align MLLM features to the dimension of T5.

**T5-only (baseline).**   We directly feed $E_{t5}$ into the editor as the instruction representation and fine-tune the editor exclusively using these embeddings. This approach serves as both our baseline and primary fine-tuning model, especially effective for handling complex instructions.

**MLLM projection.**   We encode the instruction once using Qwen2.5-VL-7B and project its embeddings to match T5 dimensions: $\hat{E}_{mllm} = E_{mllm} \times W \in \mathbb{R}^{L_m \times d}$. These projected tokens replace T5 embeddings to test the standalone encoding capability of MLLM.

**MLLM projection + T5 concatenation.**   We concatenate T5 and projected MLLM tokens along the token dimension: $\tilde{E} = [E_{t5}; \hat{E}_{mllm}] \in \mathbb{R}^{(L_{t5} + L_m) \times d}$, and add a small learned type embedding to differentiate their sources. This evaluates if T5 and MLLM embeddings complement each other.

**MLLM MetaQuery projection.**   Following the MetaQuery approach (Pan et al., 2025; Wei et al., 2025b), we append $N = 256$ special query tokens to the instruction and run a single forward pass through Qwen2.5-VL-7B. We retain only the embeddings corresponding to these query tokens, project them to dimension $d$ via $W$, and use the resulting compact representation $E_{mq} \in \mathbb{R}^{N \times d}$ for conditioning. This approach summarizes instructions into fixed-length embeddings independent of their original lengths.

## 4  EXPERIMENTS

### 4.1  EXPERIMENTAL SETUP

**Models.**   Our primary model (referred to as *Ours*) is built upon the state-of-the-art *FluxKontext dev* (Labs et al., 2025), utilizing token-wise conditioning for enhanced semantic alignment and editing robustness. For comparative ablation studies, we evaluate two additional architectures: the SD3 InstructPix2Pix model, based on *SD3-Medium* (Esser et al., 2024), which employs channel-wise conditioning, and the Flux with SigLIP model, based on *Flux 1.0 dev* (Labs, 2024), leveraging token-wise control using SigLIP features (Zhai et al., 2023). Details shown in the Appendix F.

**Benchmarks.**   We conduct comprehensive evaluations using multiple benchmarks designed to measure diverse editing capabilities. Specifically, we assess general editing performance using *GEdit-EN-full* (Liu et al., 2025) and *ImgEdit-Full* (Ye et al., 2025), and examine compositional understanding with the *Complex-Edit* benchmark (Yang et al., 2025). Full descriptions of these benchmarks are provided in the Appendix E.

### 4.2  MAIN RESULTS

As demonstrated in Tables 1, 2, and 3, our model, trained on the GPT-IMAGE-EDIT-1.5M dataset, achieves competitive performance among open-source methods and is highly competitive with leading proprietary models such as GPT-Image-1.

**GEdit-EN-full.**   Our model achieves an average score of **7.12**, surpassing all open-source models except Qwen-Image (Wu et al., 2025a). Notably, our model comprises only 12 billion parameters, representing less than half the parameter size of Qwen-Image (20+7B). When applying *thinking-rewritten* prompts—a structured clarification of ambiguous instructions using GPT-5 at inference without altering image content or evaluation metrics (examples detailed in Appendix D)—the performance further improves significantly to **7.66**, matching proprietary methods like GPT-Image-1. The improvements under rewritten prompts are particularly prominent in categories like *Motion* ($6.29 \to 7.73$, +1.44) and *Remove* ($7.17 \to 8.42$, +1.25), highlighting the model's capacity for nuanced semantic understanding and precision.

**ImgEdit-Full.**   On this benchmark, our method obtains an overall score of **3.85**, outperforming existing open-source methods except Qwen-Image, again despite having significantly fewer param-

Table 1: Comparison on the GEdit-EN-full benchmark; ($\dagger$): inference with thinking-rewritten prompts. ($\ddagger$): rewrite by Qwen3VL-32b. Only models marked with an (*) are based on our own testing; others are from official sources. Evaluations were performed by GPT-4.1.

| Model | BG Change | Color Alt. | Mat. Mod. | Motion | Portrait | Style | Add | Remove | Replace | Text | Tone | Avg |
|---|---|---|---|---|---|---|---|---|---|---|---|---|
| *Open-Sourced Models* | | | | | | | | | | | | |
| AnyEdit (Yu et al., 2024) | 4.31 | 4.25 | 2.64 | 0.67 | 1.90 | 1.95 | 3.72 | 3.75 | 3.23 | 0.77 | 4.21 | 2.85 |
| MagicBrush (Zhang et al., 2023) | 6.17 | 5.41 | 4.75 | 1.55 | 2.90 | 4.10 | 5.53 | 4.13 | 5.10 | 1.33 | 5.07 | 4.19 |
| Instruct-Pix2Pix (Brooks et al., 2023) | 3.94 | 5.40 | 3.52 | 1.27 | 2.62 | 4.39 | 3.07 | 1.50 | 3.48 | 1.13 | 5.10 | 3.22 |
| OmniGen (Xiao et al., 2024) | 5.23 | 5.93 | 5.44 | 3.12 | 3.17 | 4.88 | 6.33 | 6.35 | 5.34 | 4.31 | 4.96 | 5.01 |
| Step1X-Edit (Liu et al., 2025) | 7.03 | 6.26 | 6.46 | 3.66 | 5.23 | 7.24 | 7.17 | 6.42 | 7.39 | 7.40 | 6.62 | 6.44 |
| Bagel (Deng et al., 2025) | 7.44 | 6.99 | 6.26 | 5.09 | 4.82 | 6.04 | 7.94 | 7.37 | 7.31 | 7.16 | 6.17 | 6.60 |
| Bagel-thinking (Deng et al., 2025) | 7.22 | 7.24 | 6.69 | 7.12 | 6.03 | 6.17 | 7.93 | 7.44 | 7.45 | 3.61 | 6.36 | 6.66 |
| Ovis-U1 (Wang et al., 2025a) | 7.49 | 6.88 | 6.21 | 4.79 | 5.98 | 6.46 | 7.49 | 7.25 | 7.27 | 4.48 | 6.31 | 6.42 |
| OmniGen2 (Wu et al., 2025b) | - | - | - | - | - | - | - | - | - | - | - | 6.42 |
| Step1X-Edit(v1.1) (Liu et al., 2025) | 7.45 | 7.38 | 6.95 | 4.73 | 4.70 | 7.11 | 8.20 | 7.59 | 7.80 | 7.91 | 6.85 | 6.97 |
| FluxKontext dev * (Labs et al., 2025) | 7.06 | 7.03 | 5.52 | 5.62 | 4.68 | 5.55 | 6.95 | 6.76 | 6.13 | 6.10 | 7.48 | 6.26 |
| Qwen-Image (Wu et al., 2025a) | - | - | - | - | - | - | - | - | - | - | - | 7.56 |
| *Proprietary Models* | | | | | | | | | | | | |
| Gemini | 7.11 | 7.14 | 6.47 | 5.67 | 3.99 | 4.95 | 8.12 | 6.89 | 7.41 | 6.85 | 7.01 | 6.51 |
| Doubao | 8.07 | 7.36 | 7.20 | 5.38 | 6.28 | 7.20 | 8.05 | 7.71 | 7.87 | 4.01 | 7.67 | 6.98 |
| GPT-Image-1 | 6.96 | 6.85 | 7.10 | 5.41 | 6.74 | 7.44 | 7.51 | 8.73 | 8.55 | 8.45 | 8.69 | 7.49 |
| **Ours** | 7.39 | 7.43 | 7.07 | 6.29 | 6.91 | 6.62 | 7.84 | 7.36 | 7.17 | 6.22 | 8.04 | 7.12 |
| **Ours$^\dagger$ (rewrite by GPT5)** | 7.87 | 8.02 | 7.02 | 7.73 | 7.53 | 7.05 | 8.56 | 7.78 | 8.42 | 6.21 | 8.02 | 7.66 |
| Δ | (+0.48) | (+0.59) | (-0.05) | (+1.44) | (+0.62) | (+0.43) | (+0.72) | (+0.42) | (+1.25) | (-0.01) | (-0.02) | (+0.54) |
| **Ours$^\ddagger$ (rewrite by Qwen3VL-32b)** | 7.86 | 7.73 | 7.12 | 8.23 | 7.44 | 7.23 | 8.37 | 8.57 | 7.93 | 6.05 | 8.15 | 7.70 |
| Δ | (+0.47) | (+0.30) | (+0.05) | (+1.94) | (+0.53) | (+0.61) | (+0.53) | (+1.21) | (+0.76) | (-0.17) | (+0.11) | (+0.58) |

Table 2: Comparison on the ImgEdit-Full benchmark; ($\dagger$): inference with thinking-rewritten prompts. ($\ddagger$): rewrite by Qwen3VL-32b. Only models marked with (*) are based on our own testing; others are from official sources. Evaluations were performed by GPT-4.1.

| Model | Add | Adjust | Extract | Replace | Remove | Background | Style | Hybrid | Action | Overall |
|---|---|---|---|---|---|---|---|---|---|---|
| MagicBrush (Zhang et al., 2024) | 2.84 | 1.58 | 1.51 | 1.97 | 1.58 | 1.75 | 2.38 | 1.62 | 1.22 | 1.90 |
| Instruct-Pix2Pix (Brooks et al., 2023) | 2.45 | 1.83 | 1.44 | 2.01 | 1.50 | 1.44 | 3.55 | 1.20 | 1.46 | 1.88 |
| AnyEdit (Yu et al., 2024) | 3.18 | 2.95 | 1.88 | 2.47 | 2.23 | 2.24 | 2.85 | 1.56 | 2.65 | 2.45 |
| UltraEdit (Zhao et al., 2024) | 3.44 | 2.81 | 2.13 | 2.96 | 1.45 | 2.83 | 3.76 | 1.91 | 2.98 | 2.70 |
| OmniGen (Xiao et al., 2024) | 3.47 | 3.04 | 1.71 | 2.94 | 2.43 | 3.21 | 4.19 | 2.24 | 3.38 | 2.96 |
| Step1X-Edit (Liu et al., 2025) | 3.88 | 3.14 | 1.76 | 3.40 | 2.41 | 3.16 | 4.63 | 2.64 | 2.52 | 3.06 |
| ICEdit (Zhang et al., 2025) | 3.58 | 3.39 | 1.73 | 3.15 | 2.93 | 3.08 | 3.84 | 2.04 | 3.68 | 3.05 |
| BAGEL (Deng et al., 2025) | 3.56 | 3.31 | 1.70 | 3.30 | 2.62 | 3.24 | 4.49 | 2.38 | 4.17 | 3.20 |
| UniWorld-V1 (Lin et al., 2025) | 3.82 | 3.64 | 2.27 | 3.47 | 3.24 | 2.99 | 4.21 | 2.96 | 2.74 | 3.26 |
| OmniGen2 (Wu et al., 2025b) | 3.57 | 3.06 | 1.77 | 3.74 | 3.20 | 3.57 | 4.81 | 2.52 | 4.68 | 3.44 |
| Ovis-U1 (Wang et al., 2025a) | 4.13 | 3.62 | 2.98 | 4.45 | 4.06 | 4.22 | 4.69 | 3.45 | 4.61 | 4.00 |
| FluxKontext dev * (Labs et al., 2025) | 3.76 | 3.45 | 2.15 | 3.98 | 2.94 | 3.78 | 4.38 | 2.96 | 4.26 | 3.52 |
| Qwen-Image (Wu et al., 2025a) | 4.38 | 4.16 | 3.43 | 4.66 | 4.14 | 4.38 | 4.81 | 3.82 | 4.69 | 4.27 |
| GPT-Image-1 | 4.61 | 4.33 | 2.90 | 4.35 | 3.66 | 4.57 | 4.93 | 3.96 | 4.89 | 4.20 |
| **Ours** | 4.19 | 3.79 | 2.09 | 4.22 | 3.96 | 3.90 | 4.76 | 3.23 | 4.49 | 3.85 |
| **Ours$^\dagger$ (rewrite by GPT5)** | 4.07 | 3.77 | 2.75 | 4.32 | 4.04 | 3.92 | 4.79 | 3.23 | 4.23 | 3.90 |
| Δ | (-0.12) | (-0.02) | (+0.66) | (+0.10) | (+0.08) | (+0.02) | (+0.03) | (+0.00) | (-0.26) | (+0.05) |
| **Ours$^\ddagger$ (rewrite by Qwen3VL-32b)** | 4.10 | 3.72 | 3.18 | 4.31 | 3.99 | 3.97 | 4.87 | 3.59 | 4.54 | 4.03 |
| Δ | (-0.09) | (-0.07) | (+1.09) | (+0.09) | (+0.03) | (+0.07) | (+0.11) | (+0.36) | (+0.05) | (+0.18) |

Table 3: Comparison on the Complex-Edit benchmark.

| Method | IF | IP | PQ | O |
|---|---|---|---|---|
| AnyEdit (Yu et al., 2024) | 1.60 | 8.15 | 7.25 | 5.67 |
| UltraEdit (Zhao et al., 2024) | 6.56 | 5.93 | 7.29 | 6.59 |
| OmniGen (Xiao et al., 2024) | 6.25 | 6.42 | 7.54 | 6.74 |
| FluxKontext dev (Labs et al., 2025) | 8.56 | 8.39 | 8.51 | 8.49 |
| Imagen3 (Baldridge et al., 2024) | 7.56 | 6.55 | 7.67 | 7.26 |
| SeedEdit (Shi et al., 2024) | 8.49 | 6.91 | 8.74 | 8.04 |
| GPT-Image-1 | 9.29 | 7.51 | 9.47 | 8.76 |
| **Ours** | 9.20 | 8.57 | 9.14 | 8.97 |

eters. Enabling thinking-rewritten prompts further boosts performance to **3.90**, closely rivaling proprietary systems such as GPT-Image-1 (4.20). Performance gains with rewritten prompts are observed broadly across tasks including *Add*, *Replace*, *Remove*, and *Style*, underscoring the robustness of our dataset and token-wise conditioning strategy.

**Complex-Edit (C8).** On the challenging Complex-Edit benchmark, characterized by lengthy and detailed instructions without any additional rewriting, our model demonstrates strong overall performance at **8.97**, showing an impressive balance between Instruction Following (IF: 9.20), Identity

Table 4: Official GEdit-EN (SC, PQ, Overall) and ImgEdit (Overall) metrics for three backbones. For each architecture, the first row is fine-tuned on orginal dataset and the second (**Ours**) is the same backbone fine-tuned on GPT-IMAGE-EDIT-1.5M; * denotes the official FluxKontext dev. Token-wise models (Flux w/ SigLIP, FluxKontext) gain the most from our data.

| Model Arch. | GEdit-EN | | | ImgEdit |
|---|---|---|---|---|
| | SC | PQ | Overall | Overall |
| SD3 InstructPix2Pix | 4.34 | 6.14 | 3.92 | 2.54 |
| **Ours** | 4.96 | 6.46 | 4.91 | 3.32 |
| Flux with SigLIP | 4.48 | 5.51 | 4.75 | 3.00 |
| **Ours** | 5.57 | 8.00 | 5.81 | 3.49 |
| FluxKontext* | 6.98 | 7.20 | 6.26 | 3.52 |
| **Ours** | 7.63 | 7.69 | 7.12 | 3.85 |

Preservation (IP: 8.57), and Perceptual Quality (PQ: 9.14). This balanced performance significantly exceeds that of GPT-Image-1 in IP (+1.06), while closely matching it in IF and PQ metrics. The effectiveness of our conditioning strategy in preserving identity under complex, multi-step instructions is particularly evident here.

Beyond the three main editing benchmarks, we also evaluate on reasoning-centric RISEBench and knowledge-intensive KRISBench (Appendix G). Our model outperforms prior public systems across all KRISBench knowledge levels and substantially narrows the gap to proprietary editors on RISEBench, especially under thinking-rewritten prompts.

### 4.3 ABLATION STUDIES

We conduct a series of ablation studies to isolate where the gains of our system come from, focusing on conditioning mechanisms, text encoders, and instruction robustness, all evaluated with the official GEdit-EN and ImgEdit metrics.

**Conditioning Paradigm (Channel-wise vs. Token-wise)** Table 4 compares SD3 InstructPix2Pix (channel-wise conditioning) with Flux w/ SigLIP and FluxKontext (both token-wise). After fine-tuning on GPT-IMAGE-EDIT-1.5M, all three backbones improve on the official GEdit-EN and ImgEdit scores, but the gains are clearly larger for the token-wise models. In particular, FluxKontext benefits the most, achieving the strongest semantic consistency and overall scores, while SD3 remains noticeably behind even after fine-tuning. This suggests that token-level fusion is better suited to exploit our dataset, especially under realistic spatial misalignment and challenging edits.

**Impact of Text Encoder** Table 5 studies different frozen text encoders. T5 consistently provides strong and stable performance across both GEdit-EN and ImgEdit. Replacing T5 with Qwen2.5-VL generally hurts the overall scores, even when perceptual quality can slightly improve, and compact MetaQuery-style embeddings do not close this gap. Concatenating Qwen2.5-VL with T5 recovers most of the performance and can marginally help on some GEdit-EN metrics, but the improvements are small and inconsistent. With thinking-style prompt rewriting, T5 remains the most reliable choice, indicating that shallow feature fusion with MLLMs is not yet sufficient and that deeper cross-modal integration is likely needed.

**Sensitivity to Instruction Perturbations** To probe how strongly our editors rely on the instruction semantics, we take the 747 ImgEdit prompts (using GPT-rewritten instructions) and, for each, use Qwen3-VL-32B to synthesize five perturbed variants: drop_critical, drop_noncritical, drop_random, synonym_cf, and antonym_cf. Among these, 635 cases have all five variants successfully rewritten; we evaluate them with the official ImgEdit scorer and report results in Table 7. Across all three text encoders, dropping critical spans or flipping key phrases to antonyms leads to substantial degradation, whereas dropping non-critical or random spans, or replacing with synonyms, only causes small changes. This pattern indicates that our models are genuinely sensitive to the core semantics of the instruction, while remaining relatively robust to benign paraphrases and minor wording differences.

Table 5: Text-encoder ablation on FluxKontext. Columns report official GEdit-EN (SC, PQ, Overall) and ImgEdit (Overall) scores. The top block uses original benchmark prompts; the bottom block ($\dagger$) uses thinking-rewritten prompts. "Baseline" is the public FluxKontext dev.

| Text Encoder | GEdit-EN | | | ImgEdit |
| --- | --- | --- | --- | --- |
| | SC | PQ | Overall | Overall |
| Baseline | 6.98 | 7.20 | 6.26 | 3.52 |
| Qwen2.5-3B-VL-Instruct | 5.98 | 7.46 | 5.84 | 3.60 |
| Qwen2.5-7B-VL-Instruct Metaquery | 7.28 | 7.69 | 7.05 | 3.64 |
| Qwen2.5-7B-VL-Instruct | 6.08 | **7.84** | 5.89 | 3.60 |
| Qwen2.5-7B-VL-Instruct+T5 | **7.91** | 7.52 | **7.24** | 3.80 |
| T5 | 7.63 | 7.69 | 7.12 | **3.85** |
| Baseline$^\dagger$ | 7.01 | 7.15 | 6.28 | 3.64 |
| Qwen2.5-3B-VL-Instruct$^\dagger$ | 7.22 | 7.41 | 6.79 | 3.66 |
| Qwen2.5-7B-VL-Instruct Metaquery$^\dagger$ | 7.40 | 7.58 | 7.06 | 3.69 |
| Qwen2.5-7B-VL-Instruct$^\dagger$ | 7.25 | **7.81** | 6.87 | 3.61 |
| Qwen2.5-7B-VL-Instruct+T5$^\dagger$ | 8.08 | 7.57 | 7.45 | 3.89 |
| T5$^\dagger$ | **8.23** | 7.75 | **7.66** | **3.90** |

Table 6: Comparison of FluxKontext and our fine-tuned model on three benchmarks. For each benchmark, the better score per metric is in **bold**.

| Benchmark | Model | IP | IF | PQ | Overall | CLIP-I | DINO | CLIP-Out |
| --- | --- | --- | --- | --- | --- | --- | --- | --- |
| GEditEN-Full | FluxKontext | **9.38** | 7.77 | 8.19 | 8.45 | **0.929** | **0.861** | 0.275 |
| | Ours(FluxKontext) | 8.97 | **8.28** | **8.41** | **8.56** | 0.911 | 0.809 | **0.278** |
| ImgEdit-Full | FluxKontext | **9.14** | 7.79 | 8.00 | 8.31 | **0.919** | **0.746** | 0.273 |
| | Ours(FluxKontext) | 8.99 | **8.52** | **8.48** | **8.66** | 0.904 | 0.683 | **0.281** |
| ComplexEdit-C8 | FluxKontext | 8.39 | 8.56 | 8.51 | 8.49 | 0.842 | 0.59 | 0.189 |
| | Ours(FluxKontext) | **8.57** | **9.20** | **9.14** | **8.97** | **0.858** | **0.642** | **0.193** |

## 4.4 QUALITATIVE RESULTS

Fig. 4 presents qualitative editing examples produced by our FluxKontext model fine-tuned on GPT-IMAGE-EDIT-1.5M across various editing scenarios. These results clearly illustrate the model's strong capability to interpret and follow editing instructions, generating realistic outputs while effectively preserving non-target image content. Additional qualitative examples across diverse categories are provided in Appendix H.

## 5 LIMITATION

Benchmarks partially rely on MLLM-based scoring, which can be sensitive to variations in style and phrasing. Therefore, we present both original and thinking-rewritten prompt scores side-by-side to maintain transparency. Although thinking-rewritten prompts improve semantic alignment, text rendering and fine-grained facial identity preservation (IP) continue to pose significant challenges. Additionally, our experiments under a frozen-encoder setup reveal that shallow fusion approaches are insufficient, indicating a clear need for deeper cross-modal fusion techniques in future research.

## 6 CONCLUSION

In this work, we introduced GPT-IMAGE-EDIT-1.5M, a unified dataset of over 1.5 million instruction-based editing samples systematically refined from existing sources using GPT-Image-1. Our approach significantly enhances instruction-following and perceptual quality while keeping the filtering lightweight, so the dataset still reflects realistic identity-preservation (IP) challenges. Experiments confirmed the effectiveness of our dataset and conditioning strategies, highlighting the superiority of token-wise conditioning and the robustness of T5 embeddings under frozen-encoder conditions. By releasing GPT-IMAGE-EDIT-1.5M and corresponding models, we provide a valu-

Table 7: Instruction perturbation ablations on the ImgEdit subset (747 Fullsets; 635 with all five variants). For each text encoder, we report absolute scores and, in parentheses, the change relative to the corresponding base model (green = improvement, red = drop).

| Model | Add | Adjust | Extract | Replace | Remove | Background | Style | Hybrid | Action | Overall |
|---|---|---|---|---|---|---|---|---|---|---|
| Qwen2.5-VL-7B-Instruct | 3.93 | 3.70 | 2.20 | 3.94 | 3.98 | 3.93 | 4.78 | 3.16 | 3.74 | 3.71 |
| Drop Critical | 3.60 | 3.30 | 2.12 | 3.09 | 1.25 | 2.55 | 4.32 | 2.48 | 3.46 | 2.91 |
| Δ | (-0.33) | (-0.40) | (-0.08) | (-0.85) | (-2.73) | (-1.38) | (-0.46) | (-0.68) | (-0.28) | (-0.80) |
| Drop Noncritical | 3.82 | 3.57 | 2.25 | 3.84 | 3.95 | 3.49 | 4.65 | 3.20 | 4.15 | 3.66 |
| Δ | (-0.11) | (-0.13) | (+0.05) | (-0.10) | (-0.03) | (-0.44) | (-0.13) | (+0.04) | (+0.41) | (-0.05) |
| Drop Random | 3.90 | 3.58 | 2.25 | 3.99 | 4.06 | 3.71 | 4.66 | 3.15 | 4.01 | 3.70 |
| Δ | (-0.03) | (-0.12) | (+0.05) | (+0.05) | (+0.08) | (-0.22) | (-0.12) | (-0.01) | (+0.27) | (-0.01) |
| Synonym CF | 3.79 | 3.70 | 2.14 | 3.90 | 3.97 | 3.58 | 4.58 | 3.17 | 3.98 | 3.65 |
| Δ | (-0.14) | (0.00) | (-0.06) | (-0.04) | (-0.01) | (-0.35) | (-0.20) | (+0.01) | (+0.24) | (-0.06) |
| Antonym CF | 2.62 | 3.17 | 1.90 | 3.14 | 1.05 | 2.33 | 3.37 | 1.97 | 3.34 | 2.54 |
| Δ | (-1.31) | (-0.53) | (-0.30) | (-0.80) | (-2.93) | (-1.60) | (-1.41) | (-1.19) | (-0.40) | (-1.17) |
| Qwen2.5-VL-7B-Instruct+T5 | 4.01 | 3.80 | 2.72 | 4.26 | 4.13 | 3.88 | 4.78 | 3.33 | 4.11 | 3.89 |
| Drop Critical | 3.81 | 3.29 | 2.67 | 3.53 | 1.43 | 2.91 | 4.53 | 2.45 | 3.08 | 3.08 |
| Δ | (-0.20) | (-0.51) | (-0.05) | (-0.73) | (-2.70) | (-0.97) | (-0.25) | (-0.88) | (-1.03) | (-0.81) |
| Drop Noncritical | 3.95 | 3.50 | 2.89 | 4.14 | 3.96 | 3.84 | 4.77 | 3.33 | 4.48 | 3.87 |
| Δ | (-0.06) | (-0.30) | (+0.17) | (-0.12) | (-0.17) | (-0.04) | (-0.01) | (0.00) | (+0.37) | (-0.02) |
| Drop Random | 3.88 | 3.59 | 2.99 | 4.16 | 3.99 | 3.87 | 4.87 | 3.38 | 4.41 | 3.90 |
| Δ | (-0.13) | (-0.21) | (+0.27) | (-0.10) | (-0.14) | (-0.01) | (+0.09) | (+0.05) | (+0.30) | (+0.01) |
| Synonym CF | 3.91 | 3.51 | 2.45 | 4.19 | 4.05 | 3.84 | 4.80 | 3.51 | 4.37 | 3.85 |
| Δ | (-0.10) | (-0.29) | (-0.27) | (-0.07) | (-0.08) | (-0.04) | (+0.02) | (+0.18) | (+0.26) | (-0.04) |
| Antonym CF | 2.92 | 3.20 | 1.99 | 2.82 | 1.10 | 2.28 | 3.42 | 2.29 | 3.27 | 2.59 |
| Δ | (-1.09) | (-0.60) | (-0.73) | (-1.44) | (-3.03) | (-1.60) | (-1.36) | (-1.04) | (-0.84) | (-1.30) |
| T5 | 4.05 | 3.72 | 3.18 | 4.27 | 3.99 | 3.95 | 4.85 | 3.59 | 4.54 | 3.91 |
| Drop Critical | 3.94 | 3.43 | 2.91 | 3.61 | 1.59 | 2.83 | 4.66 | 2.68 | 3.49 | 3.14 |
| Δ | (-0.11) | (-0.29) | (-0.27) | (-0.66) | (-2.40) | (-1.12) | (-0.19) | (-0.91) | (-1.05) | (-0.77) |
| Drop Noncritical | 4.03 | 3.78 | 3.03 | 4.26 | 3.86 | 3.90 | 4.79 | 3.35 | 4.59 | 3.86 |
| Δ | (-0.02) | (+0.06) | (-0.15) | (-0.01) | (-0.13) | (-0.05) | (-0.06) | (-0.24) | (+0.05) | (-0.05) |
| Drop Random | 4.11 | 3.84 | 3.10 | 4.28 | 4.03 | 3.91 | 4.80 | 3.30 | 4.59 | 3.92 |
| Δ | (+0.06) | (+0.12) | (-0.08) | (+0.01) | (+0.04) | (-0.04) | (-0.05) | (-0.29) | (+0.05) | (+0.01) |
| Synonym CF | 4.09 | 3.83 | 2.96 | 4.27 | 4.00 | 3.87 | 4.80 | 3.39 | 4.51 | 3.87 |
| Δ | (+0.04) | (+0.11) | (-0.22) | (0.00) | (+0.01) | (-0.08) | (-0.05) | (-0.20) | (-0.03) | (-0.04) |
| Antonym CF | 2.67 | 3.07 | 1.94 | 2.59 | 1.02 | 2.32 | 3.51 | 2.42 | 3.31 | 2.42 |
| Δ | (-1.38) | (-0.65) | (-1.24) | (-1.68) | (-2.97) | (-1.63) | (-1.34) | (-1.17) | (-1.23) | (-1.49) |

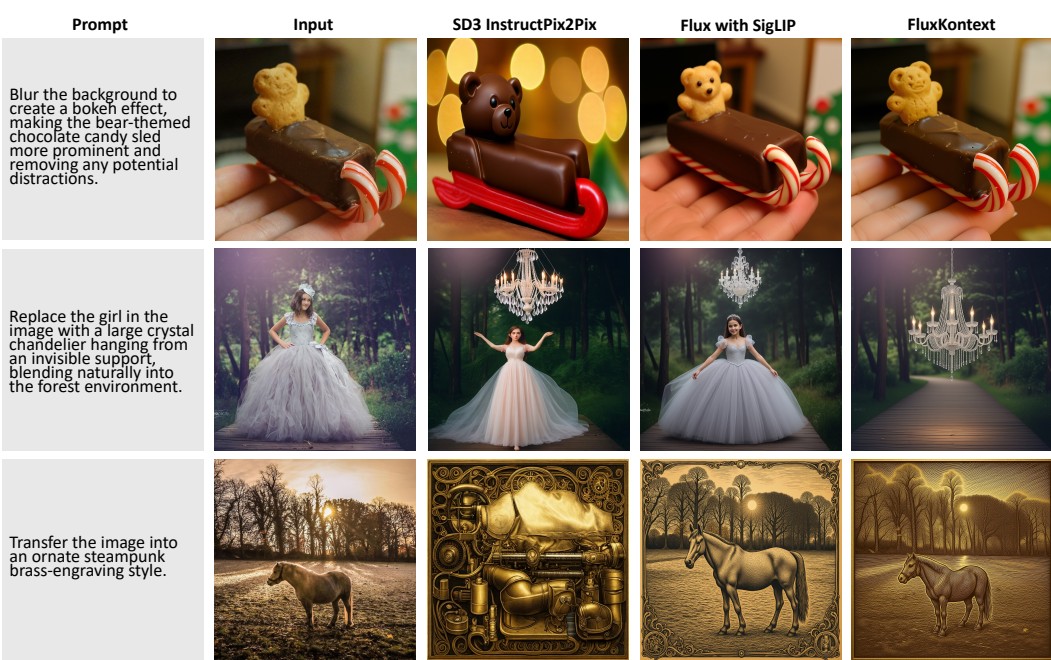

Figure 4: Qualitative comparison of editing performance across models.

able resource to accelerate future research. Future work includes exploring datasets with improved IP quality (e.g., Nano-Banana) and deeper multimodal fusion between MLLMs and MMDiT architectures.

# 7 ETHICS STATEMENT

All authors of this work have read and commit to adhering to the ICLR Code of Ethics.

# 8 REPRODUCIBILITY

To ensure reproducibility, we provide full code in the *Supplementary Material*.

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

# A  THE USE OF LARGE LANGUAGE MODELS (LLM)

During manuscript preparation, we used OpenAI GPT-4.1 for minor language refinement and writing polish. Additionally, GPT-4.1 was utilized for evaluating benchmark results. The primary dataset was generated using GPT-Image-1, with a subset of instruction prompts rewritten by GPT-4o and GPT-5. These uses of LLMs are clearly described and marked throughout the manuscript to ensure transparency.

# B  DATASET-SPECIFIC PROCESSING DETAILS

Given the heterogeneity of aspect ratios across the original datasets and the restriction that our generative model supports only three predefined ratios (1:1, 3:2, and 2:3), we adopted a standardized padding and cropping approach. Rather than directly resizing, which could distort the original content, we applied padding to each input image to match the nearest supported aspect ratio, conducted the image generation, and subsequently cropped out the padding. This process preserved the original geometry and pixel density, ensuring consistency and comparability across all datasets.

## B.1  IMPLEMENTATION DETAILS OF GPT-IMAGE-1 REGENERATION

**Per-sample regeneration pipeline.**  For each triplet in OmniEdit, HQEdit and UltraEdit we resynthesize the edited output using GPT-Image-1, while preserving the original input resolution and aspect ratio. The complete procedure is fully scriptable and resumable:

1. **Scan raw data and metadata.** We iterate over all tasks in the raw directories and, for each image $I_{\text{in}}$, read the corresponding JSON file that contains the edit history. The final instruction in `edited_prompt_list` is taken as the target editing prompt $p$. If a regenerated PNG already exists in the target folder, the sample is skipped so that the script can be safely resumed.

2. **Quantize aspect ratio.** GPT-Image-1 only accepts three canonical aspect ratios. For an input image with width $w$ and height $h$ we compute its ratio $r = w/h$ and choose the closest element from

$$\mathcal{R} = \{\texttt{square} = 1{:}1, \ \texttt{landscape} = 3{:}2, \ \texttt{portrait} = 2{:}3\},$$

by minimizing $|r - r'|$ over $r' \in \mathcal{R}$. The selected ratio determines the API resolution:

| Name | Aspect ratio $w{:}h$ | GPT-Image-1 size |
|------|---------------------|------------------|
| square | 1:1 | $1024 \times 1024$ |
| landscape | 3:2 | $1536 \times 1024$ |
| portrait | 2:3 | $1024 \times 1536$ |

3. **Pad to the canonical canvas.** We avoid direct anisotropic resizing, which would distort geometry. Instead, we pad the image to the chosen ratio while keeping all original pixels unchanged:

   • Given the target aspect ratio $r^\star$, we compute

$$(w^\star, h^\star) = \begin{cases} (\lfloor h \cdot r^\star \rfloor, \ h), & \text{if } w/h < r^\star \\ (w, \ \lfloor w/r^\star \rfloor), & \text{otherwise.} \end{cases}$$

   • We then pad on the right and/or bottom edges from $(w, h)$ to $(w^\star, h^\star)$ using a constant white color $(255, 255, 255)$.

   This produces a padded image $\tilde{I}_{\text{in}}$ whose content is geometrically identical to $I_{\text{in}}$ and whose aspect ratio is exactly one of the three supported ratios.

4. **GPT-Image-1 editing.** The padded input $\tilde{I}_{\text{in}}$ and instruction $p$ are sent to GPT-Image-1 with `quality = "high"` and the size determined in Step 2. The API returns a single edited image $\tilde{I}_{\text{out}}$ encoded as base64; we decode it into RGB space without any additional post-processing.

5. **Crop back to the original aspect ratio.** To restore the original field of view, we remove the padding before resizing:

   - Let $r_{\text{orig}} = w/h$ be the original ratio. We crop $\tilde{I}_{\text{out}}$ to the largest sub-rectangle with aspect ratio $r_{\text{orig}}$, anchored at the top-left corner:

$$(w', h') = \begin{cases} (\lfloor h^\star \cdot r_{\text{orig}} \rfloor, \ h^\star), & \text{if } w^\star/h^\star > r_{\text{orig}} \\ (w^\star, \ \lfloor w^\star/r_{\text{orig}} \rfloor), & \text{otherwise.} \end{cases}$$

   - The cropped region is then bicubically resized to exactly $(w, h)$, yielding the final edited image $I_{\text{out}}$ that is pixel-aligned with $I_{\text{in}}$.

   For reproducibility, we store both the raw GPT output $\tilde{I}_{\text{out}}$ and the cropped–resized version $I_{\text{out}}$.

**White-border and failure filtering.** Padding and internal resizing behaviours of GPT-Image-1 occasionally produce residual white borders or inconsistent crops even after the procedure above. To avoid such artefacts leaking into GPT-IMAGE-EDIT-1.5M, we apply an automatic geometric filter to $I_{\text{out}}$:

   - We first verify that the recovered resolution exactly matches the original $(w, h)$; any mismatch in width or height immediately marks the sample as a failure.

   - We then inspect a thin band along each side of the image (we use a band width of 1% of the shorter side). For every pixel in this band we compute its maximum channel distance to pure white:

$$\Delta_{\max}(x,y) = \max_{c \in \{R,G,B\}} |I_{\text{out}}^{(c)}(x,y) - 255|.$$

   Pixels with $\Delta_{\max}(x,y) \leq \tau$ (we use $\tau = 5$) are treated as "padding-colored". If the total number of such pixels exceeds $0.5\%$ of all image pixels, we regard the sample as containing visible padding or blank margins and discard it.

   - In practice, this rule removes roughly 10% of GPT-Image-1 generations, dominated by cases where the regenerated image keeps a visible white frame or where the effective crop does not fully cover the content region.

Qualitative examples of this regeneration pipeline are shown in Fig. 5. Samples that pass all checks are kept in GPT-IMAGE-EDIT-1.5M. This procedure standardizes aspect ratios without distorting the original content, while automatically filtering the most common hard failures (white borders, mis-cropped views) introduced by the regeneration step.

## B.2 ULTRAEDIT DATASET PROCESSING

The UltraEdit dataset originally provides images at a $512 \times 512$ resolution. To enhance visual fidelity and maintain benchmark compatibility, we regenerated these images at a higher resolution of $1024 \times 1024$. Afterward, bicubic interpolation was employed to downscale the regenerated images back to the original size of $512 \times 512$. This method retains high-frequency details that might otherwise be lost through direct generation at a lower resolution.

## B.3 OMNIEDIT DATASET ENHANCEMENT

For OmniEdit, additional refinements were introduced to improve data quality. Following the standard padding and cropping procedure, we systematically regenerated the output images to enhance both their visual quality and their alignment with the associated textual instructions. Recognizing semantic inconsistencies, approximately 10% of the original textual prompts were rewritten by GPT-4o using the alignment prompt in Appendix D.2, with manual spot-checking to better reflect the corresponding images. Furthermore, we augmented a substantial portion of this dataset by introducing compositional edits involving multiple sequential editing instructions, thus significantly enriching the dataset's instructional complexity.

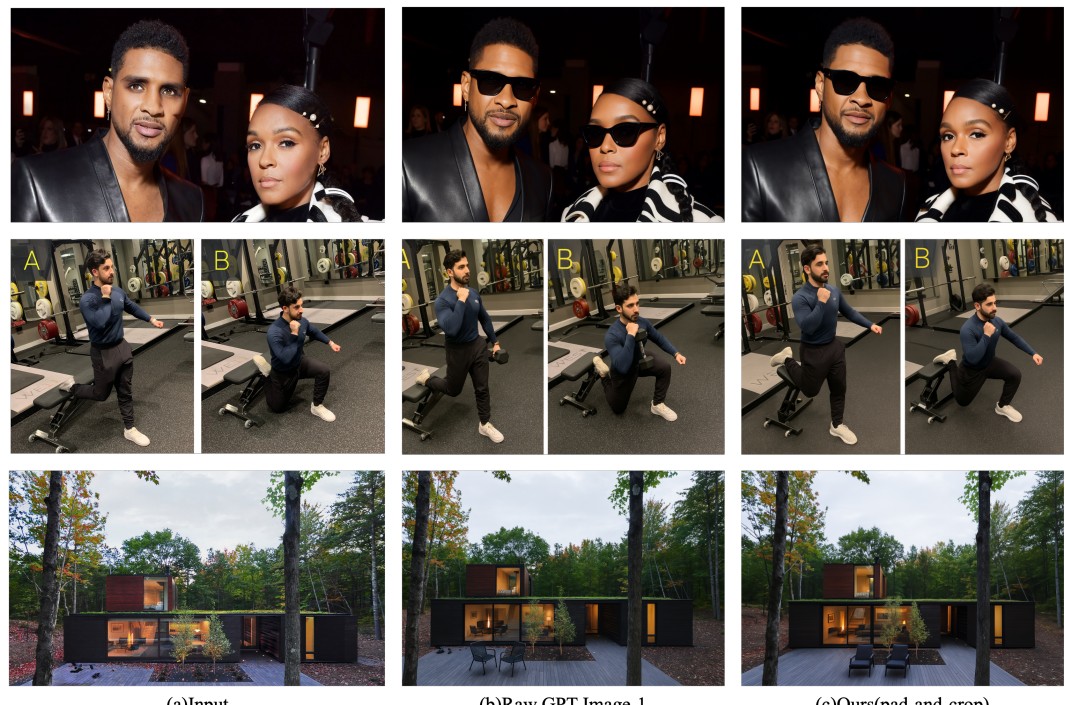

|  (a)Input | (b)Raw GPT-Image-1 | (c)Ours(pad-and-crop) |

Figure 5: Qualitative illustration of our GPT-Image-1 regeneration pipeline. For each scene (rows: portraits, human action, architecture), we show the original input image (left), a direct GPT-Image-1 edit without any aspect-ratio handling (middle), and our regenerated output using the quantized aspect ratio, pad-and-crop procedure, and crop-back step (right). Naively calling GPT-Image-1 often changes the framing or distorts the composition, whereas our pipeline preserves the original geometry and field of view while standardizing the aspect ratio.

### B.4 Complex-Edit Subset Construction

The Complex-Edit subset specifically emphasizes the dataset's compositional complexity and rigorously tests instruction-following capabilities. After applying the standard geometric preprocessing, we crafted multi-step editing instructions leveraging GPT-4o's generative capabilities. These detailed instructions, often involving two to three distinct editing operations, were subsequently used to guide GPT-Image-1 image generation. Post-generation, a rigorous filtering process eliminated any samples with detectable padding artifacts to ensure the dataset's integrity and quality.

### B.5 HQEdit Dataset: Dual-Split Strategy

The HQEdit dataset was strategically divided into two complementary subsets to maximize utility and diversity:

**Edit Split**: Existing input-instruction pairs underwent standard padding and cropping before image generation. This preserved the fidelity of original pairs while ensuring aspect ratio consistency.

**Generate Split**: For this subset, entirely new reference input images were synthesized directly from textual prompts. These generated inputs subsequently underwent editing based on the original textual instructions. To further diversify this split, aspect ratios were randomly selected from the three available presets (1:1, 3:2, 2:3), promoting variety within the generated data.

### B.6 Quality Control and Filtering Mechanisms

To uphold dataset quality, strict filtering criteria were applied after the cropping step. Samples exhibiting residual padding exceeding 0.5% of the image boundary, indicative of processing inaccuracies, were automatically excluded. Additionally, any mismatch between recorded padding masks

| Dataset | License | Usage in GPT-IMAGE-EDIT-1.5M |
|---------|---------|------------------------------|
| OmniEdit | MIT | We start from the public OmniEdit image–edit pairs (OmniEdit-Filtered-1.2M). We keep the original inputs and instructions, regenerate edited images with GPT-Image-1, and optionally apply instruction rewriting for a subset ("omniedit-gpt" / "omniedit-gpt-rewrite") to improve instruction following while preserving challenging identity-preservation (IP) cases. |
| HQ-Edit | CC BY-NC 4.0 | We reuse the HQ-Edit inputs and instructions under their non-commercial terms. For each sample, we (i) regenerate the edited image with GPT-Image-1 and (ii) optionally regenerate the full pair ("hqedit-output-regen" / "hqedit-pair-regen"), producing more consistent, high-quality triplets. |
| UltraEdit | CC BY 4.0 | We use UltraEdit as an additional source of diverse edits. We keep the original images and instructions and regenerate the edited outputs with GPT-Image-1 ("ultraedit-gpt") to unify visual quality. |

Table 8: **Source datasets and licensing.** All three sources are publicly released under research-friendly licenses (MIT / CC BY / CC BY-NC). GPT-IMAGE-EDIT-1.5M only redistributes GPT-generated edited images and associated metadata derived from these sources.

and actual cropped regions resulted in immediate rejection, thus effectively mitigating common pre-processing errors and maintaining consistent image quality standards.

### B.7 COMPREHENSIVE METADATA SCHEMA

Each dataset sample is accompanied by comprehensive JSON metadata, providing explicit documentation of data provenance and preprocessing details. This metadata encompasses source identification, original and adjusted dimensions, exact padding and cropping parameters, original and rewritten prompts (when applicable), and generation resolution specifics. Such detailed metadata supports reproducibility and offers extensive flexibility for downstream research and analysis.

## C DATA SOURCES, LICENSING, AND USAGE

**Source datasets.** GPT-IMAGE-EDIT-1.5M is constructed by systematically unifying three public instruction-based editing corpora: OmniEdit (Wei et al., 2025a), HQ-Edit (Hui et al., 2025), and UltraEdit (Zhao et al., 2024). Table 8 summarizes the licenses and our concrete usage of each source.

**License of GPT-IMAGE-EDIT-1.5M.** To harmonize the different upstream licenses and emphasize research use, we release GPT-IMAGE-EDIT-1.5M under the **Creative Commons Attribution–NonCommercial 4.0** (CC BY-NC 4.0) license.[1] This license permits copying, redistribution, and adaptation for *non-commercial* research and evaluation, provided that proper attribution to the dataset and upstream works is retained. Commercial use (including commercial fine-tuning or deployment) is explicitly excluded.

**Ownership and terms for GPT-Image-1 outputs.** All edited images in GPT-IMAGE-EDIT-1.5M are generated with OpenAI's GPT-Image-1 using prompts derived from the three source datasets. Under OpenAI's Terms of Use, as between the user and OpenAI, the user retains ownership of their Input and is assigned OpenAI's rights in the generated Output.[2] At the same time, the Terms explicitly prohibit using the Services or Output "to develop models that compete with OpenAI".[3] In our release, we (i) only redistribute GPT-Image-1 outputs that we are entitled to under these

---

[1] We will update the public dataset card accordingly.

[2] See OpenAI Terms of Use, "Ownership of content" section.

[3] See OpenAI Terms of Use, "What you cannot do" section.

terms and (ii) restrict the dataset to non-commercial research via CC BY-NC 4.0. The dataset card will prominently remind downstream users that, beyond complying with our dataset license, they must also independently ensure compliance with OpenAI's Terms of Use when using GPT-Image-1 derived content.

**Intended use and known risks.** GPT-IMAGE-EDIT-1.5M is intended for academic and non-commercial research on instruction-guided image editing, including training and evaluating diffusion/flow-matching editors, studying conditioning mechanisms, and analyzing instruction following. Because the dataset retains the hard identity-preservation and text-rendering failures, it should *not* be used in safety-critical or deceptive applications (e.g., realistic face or document manipulation) without additional safeguards. We will document typical failure modes and misuse risks in the dataset card and ethics section.

## D  PROMPT REWRITING AND GUIDANCE

This section provides the concrete prompt–engineering and implementation details for all text rewriting and caption–generation components used in our work, complementing the high–level description in Sec. 3.1 and App. B.2.[4]

### D.1  EVALUATION-TIME PROMPT REWRITING FOR BENCHMARKS

**Goal and scope.** Many benchmark instructions in GEdit-EN and ImgEdit are very short (e.g., "Change the background to a city street."), which leaves important details (what to keep, what to remove, camera and lighting constraints, etc.) underspecified. During evaluation we therefore apply an automatic *thinking-style* rewrite that maps each original benchmark pair $(I_{\text{ref}}, p_{\text{raw}})$ to a structured specification (input, edit, output). Only the rewritten edit field is then used as the instruction for image editors; metrics computed with these rewritten prompts are always marked with a dagger (†) in all tables.

**Shared system prompt.** Both GPT-5 (used for the main results) and the open-source Qwen3-VL models (32B, used for ablations and open-stack reproduction) run exactly the same system prompt, which we refer to as the *Visual Edit Prompt Rewriter*:

---

**System prompt: Visual Edit Prompt Rewriter**

```
You are a Visual Edit Prompt Rewriter.  Inputs are one or more
reference images and a raw editing instruction.  Your job:  rewrite
the task into a precise, execution-ready, low-ambiguity spec for an
image editing model (not text-to-image).
Language policy:  Output in English only.  The only exception is
text that must appear inside the image --- keep it in its original
language but wrap it in double quotes.
What to produce (exactly three fields):
- input:  a concise description of what is visibly present in
the reference image(s):  subjects, pose/action, background,
lighting, existing text/logo, salient materials/colors.  For
multi-image, enumerate each image as [IMG#] ....  - edit:  one
single, actionable sentence that states the edit to apply.  Be
specific about added / removed / modified elements, positions,
sizes, materials, colors, lighting changes, and any text to
render (quoted).  - output:  the scene after applying the edit:
what the final image should look like, preserving everything
not requested to change.  Mention composition, lighting balance,
shadows/reflections, edge blending, and consistency.
Rules:  1) Editing-only:  never propose fresh compositions; always
start from the provided image(s); final result is one image.  2)
No ratio/engine params (no width/height/steps/CFG/LoRA names,
```

---

[4]We only rewrite textual instructions; images and official benchmark protocols remain unchanged.

---

**Evaluation-time prompt rewriting examples**

**Example A (Fire Hydrant, Background Change).**

*Original benchmark prompt:* "Change the background to a city street."

*Rewritten JSON (abridged):*

`"input"`: *A red fire hydrant stands on a grassy strip between a sidewalk and a quiet residential street, with parked cars, trees, and houses in the soft overcast background.*

`"edit"`: *Replace the entire background beyond the fire hydrant with an urban city street featuring multi-story commercial buildings, asphalt road, and sidewalks, removing the trees, houses, and parked car while keeping the hydrant, lighting, and depth-of-field unchanged.*

`"output"`: *The final image shows the same hydrant in front of a busy downtown street, with consistent shadows and smooth blending between the hydrant edges and the new background.*

---

**Example B (Bear by River, Background Change).**

*Original benchmark prompt:* "Change the background to a forest."

*Rewritten* `edit` *field:*

*Replace the blurred riverside background beyond the water with a dense green forest of trees and foliage, keeping the bear and river foreground unchanged, matching the existing daylight direction and preserving shallow depth-of-field around the bear's fur and water splashes.*

Figure 6: **Evaluation-time prompt rewriting.** GPT-5 / Qwen3-VL convert short, ambiguous benchmark instructions into explicit editing commands. Only the rewritten `edit` field is used for the daggered (†) scores in Tables 1–2.

```
etc.).  3) World knowledge is allowed only to clarify styles or
domain terms; never hallucinate unseen objects, brands, or private
info.  4) If the instruction is ambiguous, make the smallest safe
assumption and keep the rest unchanged; do not invent non-visible
identities.  5) Multi-image alignment:  specify source→target
mapping and alignment (scale, camera/view, color temperature,
key light direction/intensity, shadows/reflections, occlusion
handling, edge feather in pixels).  6) Keep each field  150 words;
be concrete; avoid vague words like ''etc.''  or ''nice''.
Return ONLY a JSON object with exactly three keys:  input, edit,
output.
```

**Message construction and parsing.** For a benchmark sample with reference image path `image_path` and raw instruction $p_{raw}$, we build the chat-style input as

$$\text{messages} = [(\text{system}, \text{SYS\_PROMPT}), (\text{user}, \{\text{image} : I_{ref}, \text{ text} : p_{raw}\})].$$

The model generates a JSON object:

$$\{\text{"input"}: \text{"..."}, \text{"edit"}: \text{"..."}, \text{"output"}: \text{"..."}\}.$$

We parse this JSON and use only the `"edit"` field as the rewritten instruction $\tilde{p}_{edit}$. If JSON parsing fails (rare), the raw string is used as a fallback, and the sample is recorded in a "failed" log for later inspection.

This procedure makes the editing intent explicit without modifying images or official scoring scripts, and it is applied uniformly to all methods being compared.

### D.2 TRAINING-TIME INSTRUCTION REWRITING FOR GPT-GENERATED TRIPLETS

Beyond evaluation-time rewriting, we also rewrite a subset of training instructions to better align with GPT-Image-1 regenerated outputs (especially on OmniEdit and HQEdit).

**Inputs and outputs.** For a triplet $(I_{in}, p_{orig}, I_{out})$ produced by GPT-Image-1, we define a *data-alignment* rewriter that returns a corrected edit specification (`input, edit, output`). Here:

- `input` briefly describes the visible content of the original input $I_{in}$.

- `edit` is a single-sentence edit command that describes *only* the visual transformation from $I_{in}$ to $I_{out}$; this is the field we use for training.

- `output` summarizes the final appearance of $I_{out}$ for metadata and analysis.

We run this rewriter only on samples whose original prompts are extremely short, ambiguous, or obviously mismatched with the visual change. In practice, about 10% of OmniEdit and a smaller portion of HQEdit are rewritten in this way.

**Alignment system prompt (GPT-4o).** We use GPT-4o for the main dataset, both with a shared alignment prompt:

---

**System prompt: Visual Edit Alignment Rewriter**

```
You are an Alignment Rewriter for image editing triplets.
Inputs:  - ONE original input image (before editing).  - ONE edited
output image (after editing).  - ONE raw edit instruction text from
the dataset.
Your job:  rewrite the instruction so that it exactly describes the
visual transformation from the input image to the output image.
Produce a JSON object with three fields:
- input:  1--2 sentences summarizing what is visible in the input
image only (subjects, layout, background, lighting, any existing
text).  - edit:  ONE imperative sentence that tells an editor how
to transform the input into the output.  Mention all major changes
(added/removed/replaced objects, color/ material/pose/background
changes, text changes) but do NOT re-describe unchanged parts.
- output:  1--3 sentences describing the final edited image as
actually seen in the output.
Rules:  - Editing-only:  always treat the input image as the
starting point.  - Grounding:  rely on the two images first;
correct mistakes in the raw instruction so that edit matches the
actual visual difference.  - No hallucinations:  do NOT introduce
objects, identities, or text that are absent from the output image.
- If the raw instruction mentions an edit that did NOT happen,
quietly drop or correct it.  - Language:  same as the original
instruction (English in our datasets).
Return ONLY valid JSON with keys:  input, edit, output.
```

---

**Example (OmniEdit-style).** *Original triplet:* the raw instruction says "Turn this photo into a pop art piece.", but the output image also changes the background to a bright gradient and increases saturation of the subject.

*Rewritten* `edit`*: Convert the photo into a high-contrast pop-art style with bold outlines, neon color blocks, and a radial gradient background, while keeping the main subject's pose and composition unchanged.*

This corrected edit prompt is used during training instead of the original ambiguous one.

## D.3 CAPTIONING EDITED OUTPUTS

For retrieval, visualization, and some analysis tasks we also attach a short, human-style caption to each edited result. Captions are *not* used as supervision for our editors; they serve purely as metadata.

**Caption generator.** We use Qwen3-VL-32B with the following captioning prompt:

**System prompt: Edited-Image Captioner**

```
You are an image-to-caption generator for edited outcomes.
Task:  Given ONE input image and ONE edit instruction, write
exactly ONE short caption that describes the FINAL image AFTER the
edit has been applied.
Hard rules:  - Max 15 words (prefer 5--12).  - Output ONLY the
caption text.  One line.  No quotes.  No punctuation.  No hashtags.
- Describe the final scene; NEVER mention editing actions (no
"change", "add", "remove").  - Ground details in the provided
input image and the edit instruction.  - If the instruction adds
or replaces content, briefly mention the new object or attribute.
- If the instruction removes content, describe what remains; do
not mention the removal.  - For attribute changes, mention the NEW
attribute.  - Avoid subjective adjectives and camera/style jargon
unless explicitly required.  - Language:  English.
If any rule is broken, rewrite the caption to satisfy all rules.
```

Given a record with image $I_{\text{in}}$ and edit instruction $p$, the captioner generates a single line $c$ describing the expected final image. We post-process $c$ by trimming quotes, removing trailing punctuation, and truncating to at most 15 tokens.

## D.4 COUNTERFACTUAL INSTRUCTIONS AND TOKEN-DROP ABLATIONS

To probe how different conditioning strategies and text encoders react to perturbations in the edit instruction, we generate controlled families of counterfactual prompts for a subset of ImgEdit instructions.

**Variant types.** For each original instruction $p_{\text{orig}}$, we ask Qwen3-VL-32B (with image context) to produce five variants:

- `drop_critical`: remove the main edit operation or its core semantic span.
- `drop_noncritical`: keep the main edit unchanged, but drop stylistic or secondary modifiers.
- `drop_random`: delete a random non-core content word or short phrase.
- `synonym_cf`: replace the main edit verb or key attribute with a close synonym.
- `antonym_cf`: replace the main edit verb or attribute with a clear antonym (opposite operation).

**Variant system prompt.** All variants are produced with a single instruction-perturbation prompt:

**System prompt: Instruction Variant Generator**

```
You are an instruction rewriting assistant for an image editing
experiment.
Given ONE image and ONE edit instruction, output a JSON object with
five fields:  { "drop_critical", "drop_noncritical", "drop_random",
"synonym_cf", "antonym_cf" }, each being a single sentence in the
same language and style as the original.
Use the image only to decide which phrases are critical versus
non-critical; never invent content that contradicts the image.
Work at the span level (1--5 words).  Do not mention that you
are dropping or replacing tokens; just output the final rewritten
instructions.
Return ONLY valid JSON with those five keys.
```

We then measure performance changes when editing models are conditioned on each variant instead of $p_{\text{orig}}$, which provides a fine-grained view of text sensitivity: `drop_critical` and `antonym_cf` particularly stress instruction grounding, while `drop_noncritical` and `synonym_cf` test robustness to benign paraphrases.

|  | Stage 1: Connector pre-training | Stage 2: FluxKontext fine-tuning |
| --- | --- | --- |
| Base editor backbone | FLUX.1-Kontext-dev (frozen) | FLUX.1-Kontext-dev (trainable) |
| Text encoders | Qwen2.5-VL-7B-Instruct | Qwen2.5-VL-7B-Instruct |
| Input resolution | $512 \times 512$ | $1024 \times 1024$ |
| Objective | Connector | Full Finetuning |
| Optimizer | Prodigy | AdamW |
| Learning rate | 1 | $1 \times 10^{-6}$ |
| Training steps | 100k steps | 100k steps |
| Batch size / GPU | 2 | 2 |
| Global batch size | $8\times$ (per-GPU batch) | $8\times$ (per-GPU batch) |
| Hardware | $8\times$A100 (80GB), mixed precision | $8\times$A100 (80GB), mixed precision |
| Approx. GPU hours (Stage 1+2 total) | $\approx 1500$ GPU hours on $8\times$A100-80GB | |

Table 9: **Training configuration and compute for GPT-Image-Edit.** The detailed hyperparameters (e.g., batch size, weight decay, gradient accumulation) are provided in our code release.

# E   BENCHMARKS AND EVALUATION PROTOCOLS

We comprehensively evaluate our approach using three established instruction-guided image editing benchmarks—*GEdit-EN (Full)*, *ImgEdit (Full)*, and *Complex-Edit*—selected for their complementary assessment of general editing quality, diverse editing operations, and compositional reasoning.

**GEdit-EN (Full).**   The GEdit-EN (Full) benchmark encompasses 11 distinct editing categories: Background Change, Color Alteration, Material Modification, Motion, Portrait, Style, Add, Remove, Replace, Text, and Tone. This categorization provides broad coverage of common editing tasks and enables detailed, category-level analysis. Following the benchmark's standard evaluation protocol, we report scores for Semantic Consistency (SC), Perceptual Quality (PQ), and an aggregated Overall metric. Additionally, where explicitly noted with a dagger (†), results include minimal inference-time prompt rewriting to clarify ambiguous instructions, while strictly maintaining original images and official scoring procedures.

**ImgEdit (Full).**   ImgEdit (Full) comprises nine task families—Add, Adjust, Extract, Replace, Remove, Background, Style, Hybrid, and Action—designed to evaluate model performance on a range of atomic, localized editing tasks. The official evaluation protocol is consistently followed, with results reported per task family as well as an aggregate Overall score. Similar to GEdit-EN, daggered (†) results indicate the use of inference-time prompt rewriting to resolve ambiguities without altering images or the official evaluation criteria.

**Complex-Edit.**   The Complex-Edit benchmark specifically targets compositional editing through multi-step or constraint-rich instructions, emphasizing a model's ability to follow complex, structured prompts. Performance evaluation includes metrics for Instruction Following (IF), Identity Preservation (IP), Perceptual Quality (PQ), and an Overall aggregate score. Notably, evaluations on Complex-Edit strictly utilize the original, detailed instructions without any inference-time rewriting.

**Reproducibility.**   To ensure full reproducibility, we adhere rigorously to official benchmark evaluation pipelines and release all evaluation scripts and configuration details.

# F   IMPLEMENTATION AND TRAINING DETAILS

## F.1   TRAINING CONFIGURATION AND COMPUTE BUDGET

We follow a unified two-stage training scheme for our FluxKontext-based GPT-Image-Edit editor and its connector variants. Stage 1 pre-trains a Qwen2.5-VL connector, and Stage 2 jointly fine-tunes the connector and *FLUX.1-Kontext-dev* on GPT-IMAGE-EDIT-1.5M. Table 9 reports the key hyperparameters and compute for the main FluxKontext-based model; the Flux w/ SigLIP and MetaQuery variants reuse the same schedule unless otherwise noted.

**Model Variants.** We evaluate three distinct multimodal diffusion transformer (MMDiT)-based editing paradigms: (i) **SD3 InstructPix2Pix**, employing channel-wise conditioning built upon *SD3-Medium*; (ii) **Flux w/ SigLIP**, leveraging token-wise conditioning with *Flux 1.0 dev* guided by SigLIP features; and (iii) our primary model **FluxKontext dev**, utilizing token-wise conditioning with *FLUX.1-Kontext-dev*. Unless explicitly stated, all text encoders remain *frozen* throughout training. Fine-tuning involves updating only lightweight projection/fusion layers and the editor backbone during Stage 2. Benchmark protocols, data handling, and inference-time instruction rewriting are consistent with the main paper.

**SD3 InstructPix2Pix (channel-wise conditioning).** This model is trained following the original *UltraEdit* training recipe and hyperparameters (optimizer, learning rate schedule, regularization techniques), with the sole exception of utilizing our curated dataset. The training runs for **10 epochs**, thereby maintaining fidelity to the original SD3 configuration for a controlled comparative evaluation against token-wise architectures.

**Flux w/ SigLIP (token-wise conditioning, two-stage training).** We adopt a two-stage training protocol aligned with the *UniWorld* configuration:

- Stage 1: MLLM Connector Pretraining. We first pretrain a connector mapping Qwen2.5-VL embeddings into the SigLIP textual embedding space using the **Prodigy** optimizer for **100k steps**. Only the connector parameters are updated in this phase.
- Stage 2: Joint Connector and Flux Fine-tuning. We subsequently fine-tune both the connector and Flux using the **AdamW** optimizer at a learning rate of **1e-6** for **50k steps**, following the *UniWorld* fine-tuning strategy (data augmentation, batch packing, evaluation intervals), with the text encoder parameters remaining frozen.

**FluxKontext dev (token-wise conditioning, two-stage training).** For FluxKontext dev, we reuse the pretrained Stage 1 connector obtained from the Flux w/ SigLIP pipeline (trained under the configuration summarized in Tab. 9), and subsequently:

- Stage 2: Joint Connector and FluxKontext Fine-tuning. Both the reused connector and FluxKontext are jointly fine-tuned for **50k steps**, mirroring the Flux w/ SigLIP fine-tuning setup (AdamW optimizer, learning rate **1e-6**) to enable direct, controlled comparisons.

**MetaQuery Connector (Qwen2.5-VL, two-stage training).** For the MetaQuery variant, training follows a similar two-stage strategy:

- Stage 1: Connector Pretraining. We pretrain the MetaQuery connector, which compresses Qwen2.5-VL embeddings into $N = 256$ query tokens, summarizing and projecting them into the editor's embedding space.
- Stage 2: Joint Connector and FluxKontext Fine-tuning. The pretrained MetaQuery connector and FluxKontext are then jointly fine-tuned under identical conditions to FluxKontext Stage 2, isolating connector design effects (refer to Sec. 3.3 for tokenization specifics).

**Hardware and Precision Settings.** All experiments utilize **8×A100 (80GB)** GPUs in a distributed data parallel setup, employing mixed-precision training when feasible. Batch sizes and gradient accumulation steps are adjusted per architecture to fully utilize GPU memory capacity, ensuring comparable training throughput across all variants.

## G EXPANDED ABLATION STUDIES

**Conditioning and official metrics.** Table 13 groups official scores and shows that training on GPT-IMAGE-EDIT-1.5M consistently improves each backbone; gains are largest for token-wise models (FluxKontext, Flux+SigLIP), supporting the benefit of token-level fusion for real-world edits. **Text encoders.** With all encoders frozen, T5 is the most reliable choice overall (Table 14; detailed per-category trends in Tables 18–19). Qwen-VL alone underperforms on text-heavy instructions (e.g., the *Text* category), while concatenating Qwen-VL with T5 recovers most categories on GEdit-EN

Table 10: RISEBench results (higher is better). Columns are Temporal, Causal, Spatial, Logical, and Overall accuracy (%). Only **Ours** rows are from our evaluation; all other numbers are copied from the official leaderboard using the same API and scoring setup. ($^\dagger$): inference with rewritten prompts.

| Model | Temporal | Causal | Spatial | Logical | Overall |
|---|---|---|---|---|---|
| *Proprietary Models* | | | | | |
| Gemini-2.5-Flash-Image | 25.9 | 47.8 | 37.0 | 18.8 | 32.8 |
| GPT-Image-1 | 34.1 | 32.2 | 37.0 | 10.6 | 28.9 |
| Gemini-2.0-Flash-exp | 8.2 | 15.5 | 23.0 | 4.7 | 13.3 |
| Seedream-4.0 | 12.9 | 12.2 | 11.0 | 7.1 | 10.8 |
| *Public Models* | | | | | |
| BAGEL (w/ CoT) | 5.9 | 17.8 | 21.0 | 1.2 | 11.9 |
| Qwen-Image-Edit | 4.7 | 10.0 | 17.0 | 2.4 | 8.9 |
| Ovis-U1 | 1.2 | 3.3 | 4.0 | 2.4 | 2.8 |
| Step1X-Edit | 0.0 | 2.2 | 2.0 | 3.5 | 1.9 |
| BAGEL | 2.4 | 5.6 | 14.0 | 1.2 | 6.1 |
| FLUX.1-Kontext-Dev | 2.3 | 5.5 | 13.0 | 1.2 | 5.8 |
| **Ours** | 8.2 | 18.9 | 16.0 | 4.7 | 12.2 |
| **Ours$^\dagger$ (rewrite)** | 15.3 | 14.4 | 34.0 | 12.9 | 19.7 |
| $\Delta$ | (+7.1) | (-4.5) | (+18.0) | (+8.2) | (+7.5) |

and remains competitive on ImgEdit. **Data curation.** On 100k-subset studies, regenerating outputs and then aligning instructions yields sizable, additive gains across both SD3 and Flux backbones (Table 15). **Complex-Edit inclusion.** Adding the Complex-Edit subset provides modest but consistent improvements on GEdit-EN ($7.03 \rightarrow 7.24$) and ImgEdit ($3.71 \rightarrow 3.80$) averages (Tables 16–17), mainly via *Motion/Hybrid/Action* categories, while *Text/Tone* remain challenging. **OmniContext.** Results on OmniContext *SINGLE* (Table 12) show balanced PF/SC and narrow the gap to proprietary systems, corroborating the qualitative trends in Fig. 9. Unless marked with $^\dagger$, all numbers use original benchmark prompts; $^\dagger$ denotes inference-time "thinking-rewrites" that clarify under-specified prompts without altering images or scoring protocols.

**RISEBench.** RISEBench (Zhao et al., 2025) targets reasoning-informed visual editing and groups cases into four reasoning types: *Temporal*, *Causal*, *Spatial*, and *Logical* reasoning. Each score we report is the official accuracy (%) under the authors' LMM-as-a-judge protocol. In Table 10 we compare our model against proprietary and public baselines. For our model, we also evaluate with thinking-rewritten prompts: this substantially boosts spatial and logical reasoning accuracy, with a smaller trade-off on causal reasoning, leading to a clear overall gain. This pattern is consistent with the intuition that more explicit instructions help multi-step, geometry-heavy edits, but can sometimes overconstrain causal narratives.

**KRISBench.** KRISBench (Wu et al., 2025c) evaluates knowledge-based reasoning in image editing along three knowledge levels (factual, conceptual, procedural). The Factual block aggregates Attribute Perception (AP), Spatial Perception (SP), and Temporal Prediction (TP); Conceptual aggregates Social Science (SS) and Natural Science (NS); Procedural aggregates Logical Reasoning (LP) and Instruction Decomposition (ID). Each sub-metric is the average of Visual Consistency, Visual Quality, Instruction Following, and Knowledge Plausibility as defined in the benchmark, and the Overall score averages across all 22 tasks. Table 11 shows that our model already outperforms prior public systems across all three knowledge levels; applying thinking-style prompt rewriting trades a small drop in factual scores for noticeable gains in conceptual and especially procedural dimensions (notably ID), yielding a higher Overall while preserving strong visual quality.

## H  ADDITIONAL QUALITATIVE RESULTS

We provide extended visualizations spanning four representative settings: *GEdit-EN* (Fig. 7), *ImgEdit* (Fig. 8), *OmniContext* (Fig. 9), and *Complex-Edit* (Fig. 10). Across categories, the model performs localized edits while preserving non-edited content, with improved semantic alignment for motion/background/style changes and multi-step compositions. Typical failure modes include minor facial drift and imperfect text replacement, mirroring the IP and text-handling challenges discussed in the main paper.

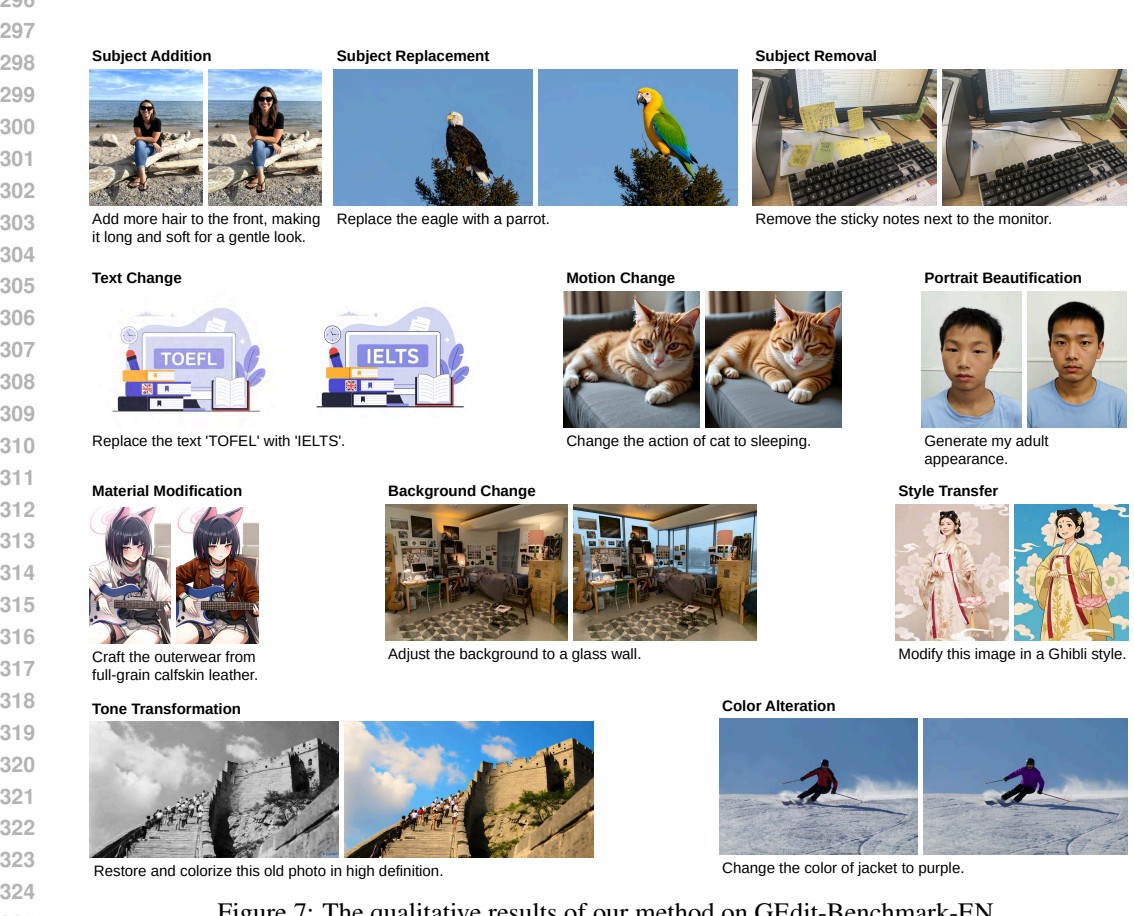

**Subject Addition**

Add more hair to the front, making it long and soft for a gentle look.

**Subject Replacement**

Replace the eagle with a parrot.

**Subject Removal**

Remove the sticky notes next to the monitor.

**Text Change**

Replace the text 'TOFEL' with 'IELTS'.

**Motion Change**

Change the action of cat to sleeping.

**Portrait Beautification**

Generate my adult appearance.

**Material Modification**

Craft the outerwear from full-grain calfskin leather.

**Background Change**

Adjust the background to a glass wall.

**Style Transfer**

Modify this image in a Ghibli style.

**Tone Transformation**

Restore and colorize this old photo in high definition.

**Color Alteration**

Change the color of jacket to purple.

Figure 7: The qualitative results of our method on GEdit-Benchmark-EN.

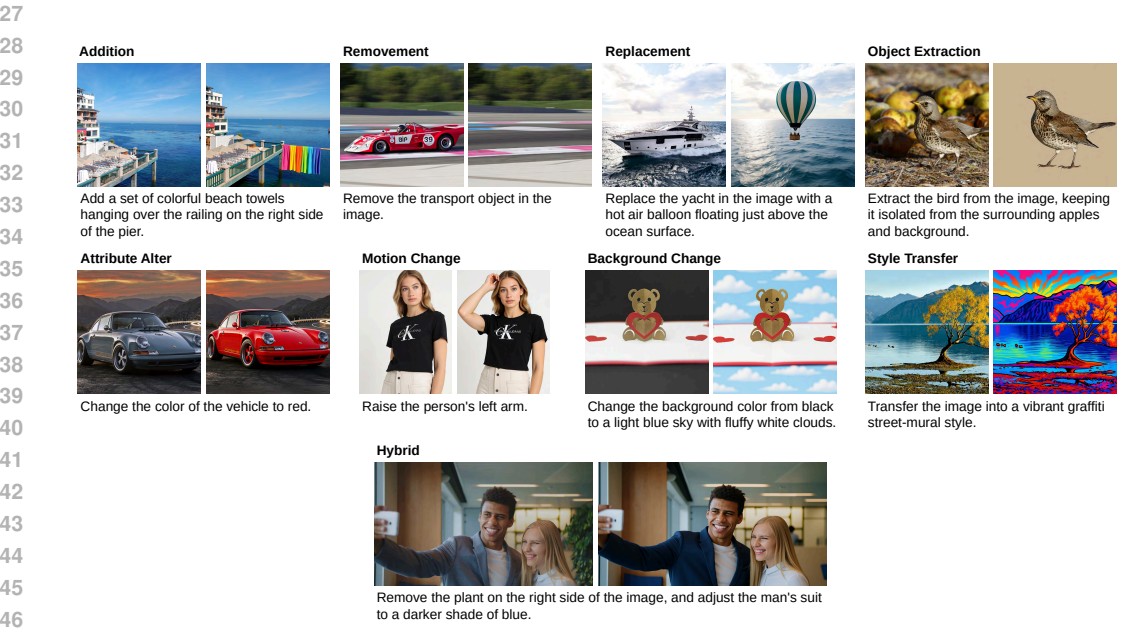

**Addition**

Add a set of colorful beach towels hanging over the railing on the right side of the pier.

**Removement**

Remove the transport object in the image.

**Replacement**

Replace the yacht in the image with a hot air balloon floating just above the ocean surface.

**Object Extraction**

Extract the bird from the image, keeping it isolated from the surrounding apples and background.

**Attribute Alter**

Change the color of the vehicle to red.

**Motion Change**

Raise the person's left arm.

**Background Change**

Change the background color from black to a light blue sky with fluffy white clouds.

**Style Transfer**

Transfer the image into a vibrant graffiti street-mural style.

**Hybrid**

Remove the plant on the right side of the image, and adjust the man's suit to a darker shade of blue.

Figure 8: The qualitative results of our method on Img-Edit.

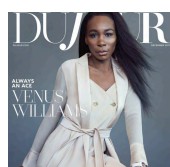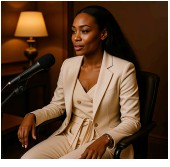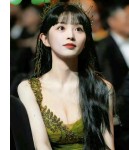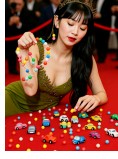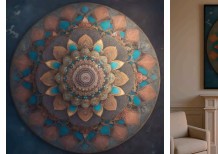

In a cozy studio, the person sits in a chair and speaks into a microphone.

A woman is playing with colorful beads and small cars on the red carpet.

Display the intricate mandala artwork on the wall above the marble fireplace in the elegant living room, enhancing the serene atmosphere as the fire crackles softly.

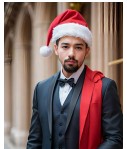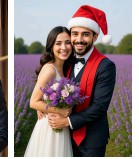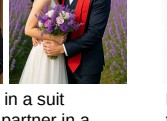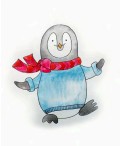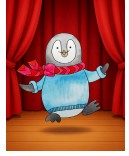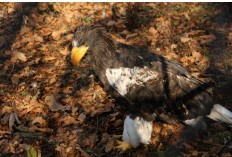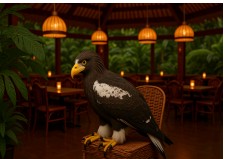

Show the man in a suit embracing his partner in a lavender field, both smiling and holding bouquets.

Dance on the stage beneath the red curtain.

In the cozy corner of a rustic tropical restaurant, a Steller's sea eagle perches gracefully on a wicker chair, its sharp eyes scanning the warm ambiance filled with greenery and soft hanging lights.

Figure 9: The qualitative results of our method on OmniContext.

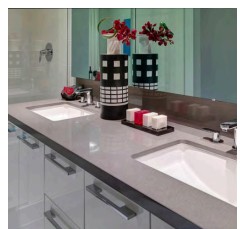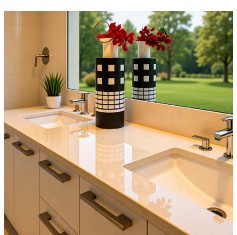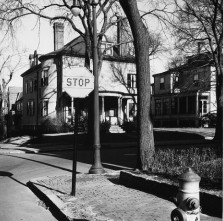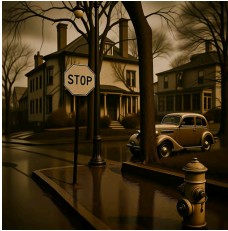

Modify the mirror reflections to display a window with a garden view and increase the ambient lighting. Replace the black faucet with a stainless-steel one, and change the countertop texture to a white marble finish. Remove the soapdish near the right sink. Resize the ornamental vase to make it slimmer, and add a small potted plant next to the left sink. Apply a warm filter to the image.

Transform the scene into a moody, rainy vintage depiction by replacing the clear sky with a cloudy, stormy one, colorizing with muted vintage tones, and adding falling raindrops. Introduce a vintage car near the curb, apply a wet, reflective texture to the road, and dim the overall lighting while adding subtle fog at ground level. Conclude with a sepia-tone filter to enhance the vintage atmosphere.

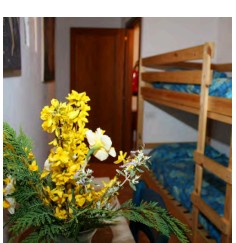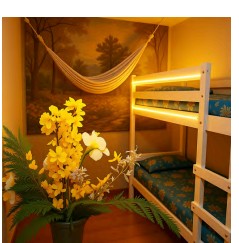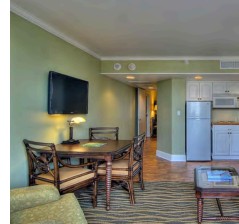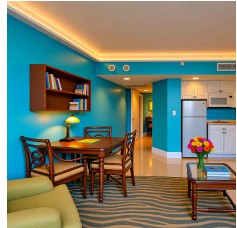

Replace the wall with an artistic mural and change the wooden bunk bed frame to white. Add a hammock hanging above the bunk beds diagonally, and modify the bedding to have floral patterns. Remove the chair near the flowers. Illuminate the scene with warm golden light, add a glowing aura to the bouquet, and finish with a vintage-style filter.

Change the wall color to a vibrant sky blue and the floor to a polished marble texture. Replace the TV with a wooden shelf with books and move the coffee table slightly closer to the couch. Add a vase with colorful flowers on the dining table, increase the brightness of the lamp, and apply a warm sepia tone across the image. Introduce softly glowing light trails along the ceiling corners.

Figure 10: The qualitative results of our method on Complex-Edit.

Table 11: KRISBench results (higher is better). We report Factual, Conceptual, and Procedural sub-metrics, plus the official Overall score. Only **Ours** rows are from our evaluation; all other numbers are taken from the official leaderboard under the same API and scoring protocol. ($^\dagger$): inference with rewritten prompts.

| Model | Factual↑ | | | | Conceptual↑ | | | Procedural↑ | | | Overall↑ |
|---|---|---|---|---|---|---|---|---|---|---|---|
| | AP | SP | TP | Avg | SS | NS | Avg | LP | ID | Avg | Overall |
| *Private Models* | | | | | | | | | | | |
| GPT-Image-1 | 83.17 | 79.08 | 68.25 | 79.80 | 85.50 | 80.06 | 81.37 | 71.56 | 85.08 | 78.32 | 80.09 |
| Gemini 2.0 | 66.33 | 63.33 | 63.92 | 65.26 | 68.19 | 56.94 | 59.65 | 54.13 | 71.67 | 62.90 | 62.41 |
| Step 3o vision | 69.67 | 61.08 | 63.25 | 66.70 | 66.88 | 60.88 | 62.32 | 49.06 | 54.92 | 51.99 | 61.43 |
| Doubao | 70.92 | 59.17 | 40.58 | 63.30 | 65.50 | 61.19 | 62.23 | 47.75 | 60.58 | 54.17 | 60.70 |
| FLUX.1 Kontext [Max] | 71.25 | 69.17 | 0.00 | 59.04 | 60.88 | 56.06 | 57.22 | 50.38 | 40.83 | 45.60 | 55.12 |
| FLUX.1 Kontext [Pro] | 69.42 | 70.17 | 0.00 | 58.14 | 55.44 | 54.94 | 55.06 | 50.12 | 43.25 | 46.69 | 54.17 |
| *Public Models* | | | | | | | | | | | |
| BAGEL-Think | 67.42 | 68.33 | 58.67 | 66.18 | 63.55 | 61.40 | 61.92 | 48.12 | 50.22 | 49.02 | 60.18 |
| BAGEL | 64.27 | 62.42 | 42.45 | 60.26 | 55.40 | 56.01 | 55.86 | 52.54 | 50.56 | 51.69 | 56.21 |
| Step1X-Edit v1.1 | 64.17 | 61.75 | 0.00 | 53.05 | 52.06 | 55.06 | 54.34 | 52.56 | 36.75 | 44.66 | 51.59 |
| UniWorld-V1 | 58.17 | 54.50 | 63.00 | 47.71 | 47.50 | 43.94 | 44.80 | 42.00 | 53.83 | 47.92 | 50.27 |
| OmniGen2 | 59.92 | 52.25 | 54.75 | 57.36 | 47.56 | 43.12 | 44.20 | 32.50 | 63.08 | 47.79 | 49.71 |
| FLUX.1 Kontext [Dev] | 64.83 | 60.92 | 0.00 | 53.28 | 48.94 | 50.81 | 50.36 | 46.06 | 39.00 | 42.53 | 49.54 |
| **Ours** | 72.94 | 67.00 | 58.00 | 65.98 | 73.59 | 71.86 | 72.72 | 60.57 | 64.49 | 62.53 | 68.98 |
| **Ours$^\dagger$ (rewrite)** | 71.09 | 63.42 | 55.52 | 63.34 | 77.11 | 75.05 | 76.08 | 60.05 | 75.67 | 67.86 | 70.85 |
| Δ | (-1.85) | (-3.58) | (-2.48) | (-2.64) | (+3.52) | (+3.19) | (+3.36) | (-0.52) | (+11.18) | (+5.33) | (+1.87) |

Table 12: Results on the OmniContext *SINGLE* benchmark.

| Method | Character | | | Object | | | Average | | |
|---|---|---|---|---|---|---|---|---|---|
| | PF | SC | Overall | PF | SC | Overall | PF | SC | Overall |
| InfiniteYou | 7.81 | 5.15 | 6.05 | — | — | — | — | — | — |
| UNO | 7.56 | 6.48 | 6.60 | 7.78 | 6.65 | 6.83 | 7.67 | 6.56 | 6.72 |
| BAGEL | 7.72 | 4.86 | 5.48 | 8.56 | 6.06 | 7.03 | 8.14 | 5.46 | 6.25 |
| OmniGen | 7.12 | 7.58 | 7.21 | 7.66 | 5.04 | 5.71 | 7.39 | 6.31 | 6.46 |
| OmniGen2 | 8.04 | 8.34 | 8.05 | 8.44 | 7.26 | 7.58 | 8.24 | 7.80 | 7.81 |
| Flux.1 Kontext (dev) | 7.70 | 8.72 | 8.07 | 8.76 | 8.22 | 8.33 | 8.23 | 8.47 | 8.20 |
| Flux.1 Kontext (max) | 7.98 | 9.24 | 8.48 | 8.78 | 8.76 | 8.68 | 8.38 | 9.00 | 8.58 |
| Gemini-2.0-Flash | 5.54 | 5.98 | 5.06 | 6.17 | 5.89 | 5.17 | 5.86 | 5.93 | 5.11 |
| GPT-Image-1 | 8.89 | 9.03 | 8.90 | 9.40 | 8.74 | 9.01 | 9.14 | 8.88 | 8.95 |
| **Ours** | 8.10 | 8.36 | 8.11 | 8.50 | 7.68 | 7.87 | 8.30 | 8.02 | 7.99 |

Table 13: Complex-Edit metrics results on GEdit-EN and ImgEdit; (*): indicates using pretrained FluxKontext weights.

| Conditioning Mechanism | GEdit-EN | | | | ImgEdit | | | |
|---|---|---|---|---|---|---|---|---|
| | IP | IF | PQ | Overall | IP | IF | PQ | Overall |
| SD3 InstructPix2Pix | 8.25 | 5.92 | 7.91 | 7.36 | 7.73 | 5.70 | 5.99 | 6.48 |
| **Ours(SD3 InstructPix2Pix)** | 5.58 | 7.12 | 7.42 | 6.71 | 6.33 | 8.01 | 7.91 | 7.42 |
| Flux with SigLIP | 5.46 | 5.76 | 7.70 | 6.30 | 6.49 | 6.30 | 8.69 | 7.16 |
| **Ours(Flux SigLIP)** | 7.40 | 6.08 | 8.94 | 7.47 | 7.73 | 6.95 | 9.20 | 7.96 |
| FluxKontext* | 9.38 | 7.77 | 8.19 | 8.45 | 9.14 | 7.79 | 8.00 | 8.31 |
| **Ours(FluxKontext)** | 8.97 | 8.28 | 8.41 | 8.56 | 8.99 | 8.52 | 8.48 | 8.66 |

Table 14: Text encoder ablation (Complex-Edit metrics); Top block: original prompts; bottom block: thinking-rewritten prompts ($^\dagger$).

| Text Encoder | GEdit-EN | | | | ImgEdit | | | |
|---|---|---|---|---|---|---|---|---|
| | IP | IF | PQ | Overall | IP | IF | PQ | Overall |
| **Baseline** | **9.38** | 7.77 | 8.19 | 8.45 | **9.14** | 7.79 | 8.00 | 8.31 |
| **Qwen2.5-7B-VL-Instruct Metaquery** | 8.34 | 8.11 | 8.17 | 8.21 | 8.64 | 7.99 | 8.39 | 8.34 |
| **Qwen2.5-7B-VL-Instruct** | 8.99 | 6.63 | **8.43** | 8.02 | 9.02 | 7.50 | **8.56** | 8.36 |
| **Qwen2.5-7B-VL-Instruct+T5** | 8.98 | **8.28** | 8.42 | **8.56** | 8.97 | 8.37 | 8.53 | 8.62 |
| **T5** | 8.99 | **8.28** | 8.42 | **8.56** | 8.99 | **8.52** | 8.48 | **8.66** |
| **Baseline**$^\dagger$ | **9.37** | 7.68 | 8.20 | 8.42 | **9.29** | 8.58 | 7.87 | 8.58 |
| **Qwen2.5-7B-VL-Instruct Metaquery**$^\dagger$ | 8.32 | 8.10 | 8.13 | 8.18 | 8.76 | 8.75 | 8.41 | 8.64 |
| **Qwen2.5-7B-VL-Instruct**$^\dagger$ | 8.85 | 7.54 | **8.40** | 8.26 | 9.08 | 8.88 | 8.58 | **8.85** |
| **MLLM+T5**$^\dagger$ | 9.01 | 8.77 | 8.26 | 8.68 | 8.92 | 7.91 | **8.60** | 8.48 |
| **T5**$^\dagger$ | 9.05 | **8.93** | 8.38 | **8.79** | 9.03 | **9.01** | 8.47 | 8.84 |

Table 15: Ablations on data curation (100k-subset runs). Regenerating outputs first (`-gpt`/`-output-regen`) provides large gains; aligning instructions (`-gpt-rewrite`/`-pair-regen`) adds further improvements. Trends hold for both channel-wise (SD3 InstructPix2Pix) and token-wise (Flux with SigLIP) backbones, supporting the two-step refinement pipeline.

| Method | Dataset Variant | ImgEdit | GEdit-EN |
|---|---|---|---|
| | *OmniEdit Ablations* | | |
| SD3 InstructPix2Pix | omniedit100k-base | 2.54 | 3.92 |
| SD3 InstructPix2Pix | omniedit100k-gpt | 3.13 | 4.91 |
| SD3 InstructPix2Pix | omniedit100k-gpt-rewrite | 3.32 | 4.89 |
| Flux with SigLIP | omniedit100k-base | 2.94 | 4.93 |
| Flux with SigLIP | omniedit100k-gpt | 3.24 | 5.98 |
| Flux with SigLIP | omniedit100k-gpt-rewrite | 3.40 | 5.88 |
| | *HQEdit Ablations* | | |
| SD3 InstructPix2Pix | hqedit100k-base | 2.19 | 2.00 |
| SD3 InstructPix2Pix | hqedit100k-output-regen | 3.02 | 4.45 |
| SD3 InstructPix2Pix | hqedit100k-pair-regen | 3.08 | 4.75 |
| Flux with SigLIP | hqedit100k-base | 3.12 | 4.34 |
| Flux with SigLIP | hqedit100k-output-regen | 3.44 | 5.67 |
| Flux with SigLIP | hqedit100k-pair-regen | 3.45 | 5.73 |
| | *Complex-Edit Instruction Ablation* | | |
| Flux with SigLIP | Complex-Edit | 2.89 | 5.39 |

Table 16: Effect of including the *Complex-Edit* subset on GEdit-EN (per-category). Inclusion yields a consistent average gain (+0.21), with the largest improvements in *Motion*, *Add*, and *Replace*; small trade-offs appear in *Tone* and *Text*, reflecting the challenge of long compositional instructions.

| Dataset | BG Change | Color Alt. | Mat. Mod. | Motion | Portrait | Style | Add | Remove | Replace | Text | Tone | Avg |
|---|---|---|---|---|---|---|---|---|---|---|---|---|
| Fluxkontext mllm+T5 w/o complex | 7.62 | 7.55 | 6.77 | 7.08 | 6.74 | 6.74 | 7.68 | 7.74 | 6.82 | 5.36 | 7.23 | 7.03 |
| Fluxkontext mllm+T5 (full) | 7.80 | 7.54 | 7.12 | 7.75 | 7.09 | 6.74 | 8.04 | 7.95 | 7.17 | 5.45 | 6.95 | 7.24 |

Table 17: Effect of including the *Complex-Edit* subset on ImgEdit (per-family). Overall improves from 3.71 to 3.80, driven by gains in *Hybrid* and *Action*, while other categories remain stable.

| Dataset | Add | Adjust | Extract | Replace | Remove | Background | Style | Hybrid | Action | Overall |
|---|---|---|---|---|---|---|---|---|---|---|
| Fluxkontext mllm+T5 w/o complex | 4.07 | 3.69 | 1.94 | 4.17 | 3.93 | 3.73 | 4.74 | 2.91 | 4.19 | 3.71 |
| Fluxkontext mllm+T5 (full) | 4.07 | 3.79 | 2.04 | 4.13 | 3.89 | 3.90 | 4.84 | 3.04 | 4.52 | 3.80 |

Table 18: Text-encoder configurations on GEdit-EN (per-category; encoders frozen). T5 is a reliable default; Qwen-VL alone weakens *Text* (1.20) and *Replace*, while concatenating Qwen-VL with T5 restores most categories and attains the best average (7.24).

| Text Encoder | BG Change | Color Alt. | Mat. Mod. | Motion | Portrait | Style | Add | Remove | Replace | Text | Tone | Avg |
|---|---|---|---|---|---|---|---|---|---|---|---|---|
| FluxKontext dev (T5) | 7.06 | 7.03 | 5.52 | 5.62 | 4.68 | 5.55 | 6.95 | 6.76 | 6.13 | 6.10 | 7.48 | 6.26 |
| Finetuned with T5 | 7.39 | 7.43 | 7.07 | 6.29 | 6.91 | 6.62 | 7.84 | 7.36 | 7.17 | 6.22 | 8.04 | 7.12 |
| Finetuned with QwenVL | 6.45 | 7.27 | 5.04 | 6.53 | 7.26 | 5.88 | 7.03 | 7.20 | 4.31 | 1.20 | 6.64 | 5.89 |
| QwenVL + T5 (Ours) | 7.80 | 7.54 | 7.12 | 7.75 | 7.09 | 6.74 | 8.04 | 7.95 | 7.17 | 5.45 | 6.95 | 7.24 |

Table 19: Text-encoder configurations on ImgEdit (per-family; encoders frozen). T5 attains the best overall (3.85); Qwen-VL+T5 is close (3.80) and strongest on *Style* and *Action*; Qwen-VL alone lags on *Extract* and *Replace*.

| Text Encoder | Add | Adjust | Extract | Replace | Remove | Background | Style | Hybrid | Action | Overall |
|---|---|---|---|---|---|---|---|---|---|---|
| FluxKontext dev (T5) | 3.76 | 3.45 | 2.15 | 3.98 | 2.94 | 3.78 | 4.38 | 2.96 | 4.26 | 3.52 |
| Finetuned with T5 | 4.19 | 3.79 | 2.09 | 4.22 | 3.96 | 3.90 | 4.76 | 3.23 | 4.49 | 3.85 |
| Finetuned with QwenVL | 3.92 | 3.58 | 1.95 | 3.62 | 3.89 | 3.72 | 4.64 | 3.22 | 3.82 | 3.60 |
| QwenVL + T5 (Ours) | 4.07 | 3.79 | 2.04 | 4.13 | 3.89 | 3.90 | 4.84 | 3.04 | 4.52 | 3.80 |

