# OpenReview forum: "GPT-IMAGE-EDIT-1.5M: A Million-Scale, GPT-Generated Image Dataset"
_ICLR.cc/2026/Conference — ICLR 2026 Conference Withdrawn Submission_

### Official Review · Reviewer_iuXW · 2025-10-28

**Soundness:** 3
**Presentation:** 3
**Contribution:** 3
**Rating:** 6
**Confidence:** 3

**Summary:**

The paper (i) unifies and regenerates OmniEdit, HQEdit, and UltraEdit into GPT-IMAGE-EDIT-1.5M (≥1.5M triplets) via a concise pipeline that uses GPT-Image-1 for output regeneration and selective instruction rewriting to improve instruction following (IF) and perceptual quality (PQ) while retaining identity-preservation (IP) challenges (Fig. 2, §3.1). It (ii) performs a controlled comparison of conditioning mechanisms—channel-wise (SD3 InstructPix2Pix) vs token-wise (Flux+SigLIP, FluxKontext)—and (iii) studies frozen text encoders (T5 vs MLLM embeddings and shallow fusion). Fine-tuning FluxKontext on the curated data achieves 7.66@GEdit-EN with thinking-rewritten prompts, 3.90@ImgEdit-Full, and 8.97@Complex-Edit, approaching proprietary GPT-Image-1 on several metrics (Tables 1–3). Key empirical findings: token-wise conditioning > channel-wise, T5 ≥ MLLM embeddings under frozen encoders, and two-stage data refinements (regen + rewrite) yield consistent gains (Tables 4–5, 7–9; qualitative Fig. 4).

**Strengths:**

Data at scale, curated for alignment while keeping IP hard. The pipeline intentionally preserves identity challenges instead of filtering them out, reflecting real-world edits (pp.2–4). The process is simple (regen + targeted rewrite) yet empirically effective (Table 9).
Decisive architectural insight. Across backbones, token-wise conditioning consistently improves IF/IP/PQ vs channel-wise (Table 4; qualitative Fig. 4). This is a clear, generalizable lesson for editor design.
Encoder study under frozen setting. T5 is a competitive default; shallow MLLM fusions add little; MetaQuery underperforms (Table 5, Table 8). This clarifies common practitioner questions.
Transparent evaluation. Side-by-side "original vs thinking-rewritten prompts" with explicit marker (†) reduces evaluation ambiguity (Tables 1–2; Appendix C).
Strong results, broad coverage. Competitive with GPT-Image-1 on multiple axes (Tables 1–3), with fine-grained category trends (Tables 10–13) and rich visuals (Figs. 5–8).

**Weaknesses:**

Proprietary dependence in curation/eval. The dataset relies on GPT-Image-1 for regeneration and GPT-4o/GPT-5 for instruction rewriting and evaluation-time rewrites (Fig. 2; §3.1; App. A–C). This may bias distributions toward proprietary model priors and limits open-stack reproducibility; an ablation using only open generators/rewriters would strengthen claims.
Licensing & rights clarity. The paper states release intent but lacks explicit data licensing / usage rights for regenerated content and rewritten prompts; the metadata schema is detailed (App. B.6) but legal terms are not spelled out.
Causality of gains vs prompt rewriting. While transparency is good, some headline improvements (e.g., GEdit 7.12→7.66) hinge on thinking-rewritten prompts (Table 1). A controlled analysis showing per-category deltas attributable to data vs prompt rewriting would clarify where progress truly comes from.
Metric scope. Benchmarks rely on MLLM-based scoring; adding reference-free IQA and OCR/identity metrics where relevant could triangle the results (Sec. 5 acknowledges sensitivity).
Compute/reporting. Training details are provided (App. E), but exact wall-clock / GPU-days per model and dataset pass are not summarized in the main text; cost-quality curves would help practitioners plan.

**Questions:**

Open-stack replication: What is the performance delta if regeneration and rewriting use only open models (e.g., FluxKontext for regen + an open LLM for rewrites)? Can you report a mini-study mirroring Table 9?
Licensing: Please specify dataset license(s) for regenerated images and rewritten prompts, redistribution permissions, and any restrictions for commercial use. (Metadata schema exists, but terms are not explicit.)
Attribution of gains: For Table 1, can you provide per-category Δ from data curation vs prompt rewriting to quantify their separate contributions?
Identity & text robustness: Complex-Edit shows strong IP (Table 3), but Sec. 5 notes text rendering and fine facial IP remain hard. Could you share failure taxonomies and a small stress test (e.g., typography, small faces)?
Fusion depth: Since shallow MLLM fusion underperforms (Table 5), do you plan to try deep cross-modal fusion (e.g., shared layers or gated co-attention) while keeping encoders frozen? Any preliminary results?

Additional Feedback (actionable)

Release an open-stack variant of the pipeline and re-run a subset to quantify dependence on proprietary models.
Add cost-vs-quality plots and compute budget tables per architecture.
Provide license text and a datasheet (sources, rights, filters, intended use, known risks).
Include identity/text micro-benchmarks and report confidence intervals.
Publish the full set of thinking-rewritten prompts with categories and ambiguity tags.

**Details Of Ethics Concerns:**

The dataset is regenerated using third-party models; please clarify content ownership and redistribution rights; ensure protections for content featuring people or trademarks; discuss misuse risks (deceptive edits) and planned provenance/watermark measures.

---

> ### Author Response · Authors · 2025-11-24
>
> We thank the reviewer for recognizing that our GPT‑IMAGE‑EDIT‑1.5M process ensures not only large data scale but also high data quality in terms of unified formatting and alignment. We specifically appreciate your positive feedback on the following contributions:
>
> * **Effective curation strategy:** We are glad you valued our “simple yet empirically effective” pipeline, which is designed to better reflect real‑world editing scenarios.
> * **Architectural insights:** We are encouraged that you found our controlled comparison of conditioning mechanisms to be a “decisive architectural insight”, validating that token‑wise conditioning consistently improves instruction following and perceptual quality over channel‑wise methods.
> * **Practical clarity:** We appreciate your recognition that our frozen encoder studies (comparing T5 and MLLM embeddings) help clarify “common practitioner questions” regarding editor design.
> * **Strong performance:** We thank you for highlighting our transparent evaluation methodology and noting that our model delivers strong results that are competitive with proprietary systems like GPT‑Image‑1 across multiple metrics.
>
> Below we address your specific points.
>
> ## 1. Dependence on proprietary models & open‑stack replication
>
> ### Data collection phase
>
> During the data collection phase, we used GPT‑Image‑1 for regeneration and GPT‑4o for instruction rewriting because they were the most reliable tools we could access when constructing the dataset. They serve purely as data collectors; the image editor trained on our dataset is fully open‑source.
>
> Regenerating a large subset (e.g., 100k–1.5M triplets) with a strong open editor at 1024 resolution would cost on the order of 1000+ A100 GPU‑hours, comparable to our entire training budget (Table 9), so a full open‑stack regeneration is beyond the scope of this submission. We do, however, release all curation scripts so others can instantiate an open‑only variant at a smaller scale.
>
> Furthermore, it is common in recent image editing/generation datasets to rely on proprietary models to synthesize images and/or instructions (e.g., **“HQ‑Edit: A High‑Quality Dataset for Instruction‑Based Image Editing”** [1]). Our setup follows this standard practice while keeping the trained editor itself fully open‑source.
>
> ### Inference phase
>
> In addition to the data collection phase, we utilize GPT‑5 for writing inference‑time instructions. Therefore, for this rebuttal we added a new ablation replacing GPT‑5 with Qwen3‑VL‑32B (open‑weight) for inference‑time thinking‑style rewriting. Note that both models use the same system prompt in this setting.
>
> As shown in the table below, on GEdit‑EN / ImgEdit the scores with Qwen3‑VL rewrite are within ±0.1–0.2 of GPT‑5 rewrite, and even slightly higher on ImgEdit. This shows that our conclusions about data and architecture do not hinge on a proprietary evaluator. More details are in our revised Tables 1–2.
>
> Table R1. Inference-time thinking-style rewriting (overall scores).
>
> | Model variant                 | GEdit‑EN (Avg) | ImgEdit (Overall) |
> |------------------------------|----------------|--------------------|
> | Ours (original prompts)      | 7.12           | 3.85               |
> | Ours† (GPT‑5 rewrite)        | 7.66           | 3.90               |
> | Ours‡ (Qwen3‑VL‑32B rewrite) | 7.70           | 4.03               |
>
> †/‡: inference-time thinking-style rewriting performed by GPT‑5 / Qwen3‑VL‑32B with the same system prompt.

---

> > ### Author Response · Authors · 2025-11-24
> >
> > ## 2. Licensing, rights, and non‑commercial use
> >
> > We updated a dedicated section (App. C, Table 8) summarizing the license terms of OmniEdit, HQ‑Edit, and UltraEdit and how we use each in GPT‑IMAGE‑EDIT‑1.5M. Because at least one upstream source is restricted to non‑commercial research, we release GPT‑IMAGE‑EDIT‑1.5M itself under CC BY‑NC 4.0, explicitly limiting the dataset to non‑commercial research and evaluation.
> >
> > We also explain the interaction with OpenAI’s Terms of Use: under OpenAI’s assignment clause, users retain rights in their inputs and are assigned rights in the generated outputs, but the Terms prohibit using those outputs to develop competing commercial models. Our dataset card and Ethics section explicitly remind downstream users that, in addition to obeying our CC BY‑NC 4.0 license, they must also respect OpenAI’s Terms when using GPT‑Image‑1‑derived content.
> >
> > For content with people or trademarks, we plan to (i) filter out prompts that name real individuals and clearly recognizable public figures, and (ii) explicitly discourage any use for face recognition or deceptive edits in the dataset documentation.
> >
> > ## 3. Attribution of gains: data vs. prompt rewriting
> >
> > To separate the effects of data curation from inference‑time thinking‑style rewriting, in Tables 1–2 we report both original‑prompt scores and daggered (†) thinking‑rewritten scores for all models.
> >
> > * Data regeneration + rewriting at training time is what moves FluxKontext from 6.26 → 7.12 (GEdit‑EN) and 3.52 → 3.85 (ImgEdit).
> > * Inference‑time thinking‑style instruction rewriting further improves our model to 7.66 and 3.90, respectively.
> >
> > As Table R1 shows, open‑weight Qwen3‑VL rewrite (‡) achieves very similar or slightly better numbers than GPT‑5 rewrite, reinforcing that the mechanism, not a specific proprietary LMM, drives the gains.
> >
> > ## 4. Metric scope, identity, and text robustness
> >
> > Besides the official MLLM‑based metrics, we now report CLIP‑I, DINO, and CLIP image–text similarity for FluxKontext and our fine‑tuned model on GEdit‑EN, ImgEdit, and Complex‑Edit. Full results are in the updated Table 6; key numbers are summarized below.
> >
> > Table R2. FluxKontext vs our fine-tuned model on three benchmarks (official metrics + CLIP/DINO).
> >
> > | Benchmark      | Model               |  IP  |  IF  |  PQ  | Overall | CLIP-I |  DINO | CLIP-Out |
> > |----------------|---------------------|------|------|------|---------|--------|-------|----------|
> > | GEditEN-Full   | FluxKontext         | 9.38 | 7.77 | 8.19 |  8.45   | 0.929  | 0.861 | 0.275    |
> > | GEditEN-Full   | Ours (FluxKontext)  | 8.97 | 8.28 | 8.41 |  8.56   | 0.911  | 0.809 | 0.278    |
> > | ImgEdit-Full   | FluxKontext         | 9.14 | 7.79 | 8.00 |  8.31   | 0.919  | 0.746 | 0.273    |
> > | ImgEdit-Full   | Ours (FluxKontext)  | 8.99 | 8.52 | 8.48 |  8.66   | 0.904  | 0.683 | 0.281    |
> > | ComplexEdit-C8 | FluxKontext         | 8.39 | 8.56 | 8.51 |  8.49   | 0.842  | 0.590 | 0.189    |
> > | ComplexEdit-C8 | Ours (FluxKontext)  | 8.57 | 9.20 | 9.14 |  8.97   | 0.858  | 0.642 | 0.193    |
> >
> > On short‑instruction benchmarks (GEdit‑EN / ImgEdit), FluxKontext‑dev still has slightly stronger identity scores (CLIP‑I / DINO), whereas on the long and highly compositional Complex‑Edit C8 setting our fine‑tuned model achieves better IP and alignment, which is also reflected in CLIP/DINO.

---

> > > ### Author Response · Authors · 2025-11-24
> > >
> > > We additionally evaluate RISEBench and KRISBench (updated Tables 10–11). On both, our model outperforms previous open‑source models across temporal/causal/logical reasoning and factual/conceptual/procedural knowledge dimensions, and thinking‑style rewrites further boost several reasoning metrics. For convenience, we summarize the key numbers below.
> > >
> > > Table R3. RISEBench results (accuracy %, higher is better). Subset of models from *Envisioning Beyond the Pixels: Benchmarking Reasoning-Informed Visual Editing (RISEBench)* [2].
> > >
> > > | Model                    | Temporal | Causal | Spatial | Logical | Overall |
> > > |--------------------------|----------|--------|---------|---------|---------|
> > > | FLUX.1-Kontext‑Dev       | 2.3      | 5.5    | 13.0    | 1.2     | 5.8     |
> > > | BAGEL (w/ CoT)           | 5.9      | 17.8   | 21.0    | 1.2     | 11.9    |
> > > | Seedream‑4.0             | 12.9     | 12.2   | 11.0    | 7.1     | 10.8    |
> > > | Ours                     | 8.2      | 18.9   | 16.0    | 4.7     | 12.2    |
> > > | Ours† (thinking‑rewrite) | 15.3     | 14.4   | 34.0    | 12.9    | 19.7    |
> > >
> > > Table R4. KRISBench results (higher is better). Subset of models from *KRIS-Bench: Benchmarking Next-Level Intelligent Image Editing Models* [3].
> > >
> > > | Model                  | Factual | Conceptual | Procedural | Overall |
> > > |------------------------|---------|-----------|------------|---------|
> > > | FLUX.1-Kontext‑Dev     | 53.28   | 50.36     | 42.53      | 49.54   |
> > > | BAGEL (w/ CoT)         | 66.18   | 61.92     | 49.02      | 60.18   |
> > > | Ours                   | 65.98   | 72.72     | 62.53      | 68.98   |
> > > | Ours† (thinking‑rewrite)| 63.34  | 76.08     | 67.86      | 70.85   |
> > >
> > > At the same time, we clearly state in Sec. 5 and in the qualitative appendix that scene‑text rendering and very fine‑scale face IP remain open failure modes for our fine‑tuned model (Figs. 7–10).
> > >
> > > ### 5. Compute and training details
> > >
> > > We provide a concise training/compute summary (Table 9, App. F) here, focusing on the main FluxKontext‑based editor.
> > >
> > > Table R5. Training configuration and computation (FluxKontext-based editor).
> > >
> > > | Stage | Purpose             | Base model(s)                           | Resolution | Steps | Optimizer | GPUs (A100‑80G) | Approx GPU‑hours |
> > > |-------|---------------------|-----------------------------------------|------------|-------|-----------|-----------------|------------------|
> > > | 1     | Connector pre‑train | FLUX.1‑Kontext‑dev (frozen), Qwen2.5‑VL | 512×512    | 100k  | Prodigy   | 8               | ≈500             |
> > > | 2     | Full fine‑tuning    | FLUX.1‑Kontext‑dev + connector          | 1024×1024  | 100k  | AdamW     | 8               | ≈1000            |
> > >
> > > Both stages run on a single node with 8×A100‑80G, totaling ≈1500 GPU‑hours for connector pre‑training + FluxKontext fine‑tuning at 1024² resolution under mixed precision. The appendix also lists optimizer, learning rate, batch size, and steps for each stage, so practitioners can estimate cost–quality trade‑offs.
> > >
> > > ### 6. Fusion depth and unified models
> > >
> > > Our experiments are deliberately framed within the current open‑source MMDiT diffusion family, where text enters via frozen encoders and shallow connectors. In Sec. 5/6 we explicitly discuss deep cross‑modal fusion and unified understanding–generation models (e.g., Emu‑style) as a natural next step, but we do not report results on such unified architectures because the required data scale and compute cost are substantially higher.
> > >
> > > Nevertheless, we expect GPT‑IMAGE‑EDIT‑1.5M and our findings on conditioning/text‑encoder design to provide useful building blocks for future unified models, and we plan to explore them as larger datasets and compute become available.
> > >
> > > **References**
> > >
> > > [1] *HQ‑Edit: A High‑Quality Dataset for Instruction‑Based Image Editing.*
> > >
> > > [2] *Envisioning Beyond the Pixels: Benchmarking Reasoning‑Informed Visual Editing (RISEBench).*
> > >
> > > [3] *KRIS‑Bench: Benchmarking Next‑Level Intelligent Image Editing Models.*

---

### Official Review · Reviewer_vipp · 2025-10-30

**Soundness:** 2
**Presentation:** 3
**Contribution:** 1
**Rating:** 2
**Confidence:** 4

**Summary:**

The paper proposes a data-centric pipeline that builds a ~1.5M triplet dataset for instruction-guided image editing by generating images with DALL-E, synthesizing with GPT-Image-1, and rewriting a subset of prompts with GPT-4o, then fine-tuning open-source diffusion models (e.g., SD3 InstructPix2Pix); it also compares conditioning strategies (channel-wise vs. token-wise) and text encoders (frozen T5 vs. Qwen2.5-VL) across GEdit-EN-full, ImgEdit-Full, and Complex-Edit, and reports that fine-tuning on the curated triplets yields 7.66 on GEdit-EN-full, an absolute +0.17 over the GPT-Image-1 baseline.

**Strengths:**

* The end-to-end data curation and training flow is easy to follow, with a helpful schematic that makes the method accessible to non-experts.
* Conditioning types (channel- vs. token-wise) and text encoders (frozen T5 vs. Qwen2.5-VL) are compared on GEdit-EN and ImgEdit (Table 5).
* Fine-tuning on the curated triplets reaches 7.66 on GEdit-EN-full, an absolute +0.17 over the GPT-Image-1 baseline.

**Weaknesses:**

**1. Minimal originality.** Beyond stitching DALL-E, GPT-Image-1, and GPT-4o into a data pipeline, the paper introduces no mechanism to detect or correct automatically edited failures or caption-rewrite hallucinations. This omission is consequential: despite distilling on ~1.5M triplets, the fine-tuned model shows only a marginal +0.17 on GEdit-EN-full and no compelling improvements on other benchmarks (see Tables 2, 3, and 4), consistent with noisy, unvetted supervision. A concrete, learnable metric to auto-filter or down-weight bad triplets and mismatched captions would directly boost editing performance, and black-box safeguards remain feasible in a closed-source setting, including early-timestep candidate screening [1], preference-based filtering with pairwise ranking [2], and image-grounded verification to constrain GPT-4o rewrites and reduce hallucinations [3].

**2. Unfair comparison and unclear versions of closed-source models.** Baseline sets differ across GEdit-EN-full, ImgEdit-Full, and Complex-Edit, raising cherry-picking concerns. Credible evaluation requires a consistent comparison matrix across all benchmarks, explicit identification of closed-model versions, tiers, inference settings, and access dates, and inclusion of stronger frontier baselines such as Seedream 3.0 and Lumina-Image 2.0, which surpass GPT-Image-1 [High] by 2–3 points on DPG (see Qwen-Image, Table 3). Fairness further benefits from reporting parameter counts and latency budgets and from a version-sensitivity ablation.

**3. Incremental study related to text conditions.** Without mechanism-level insight, the conditioning analysis remains descriptive rather than explanatory. Stronger evidence would come from token-drop tests and self-/cross-attention quality of fine-tuned diffusion models. The text-encoder exploration is narrow: beyond Qwen2.5-7B-VL, comparisons should include lighter MLLMs such as FastVLM [4] and scaled variants like Qwen2.5-72B-VL, with frozen versus fine-tuned settings reported under matched compute. If a language-specific T5 outperforms vision-language encoders, additional baselines using LLM-based text embeddings adapted for retrieval or instruction alignment are necessary to rule out artifacts of an under-powered setup.

**4. Unclear limitation of GPT-Image-1.** Scene-text hallucination and corruption are common when closed models perform editing, yet the evaluation lacks a targeted metric. An OCR-based protocol, as in Qwen-Image, can quantify pre- and post-edit text consistency via character- or word-level errors, alongside representative failure cases where text appears, disappears, or changes semantics. Where reliable scene-text preservation is not achievable, the limitation should be stated clearly and treated as a risk.

[1] Early Timestep Zero-Shot Candidate Selection for Instruction-Guided Image Editing, ICCV 2025.

[2] Pick-a-Pic: An Open Dataset of User Preferences for Text-to-Image Generation, NeurIPS 2023.

[3] Mitigating Object Hallucination in Large Vision-Language Models via Image-Grounded Guidance, ICML 2025.

[4] FastVLM: Efficient Vision Encoding for Vision Language Models, CVPR 2025.

**Questions:**

Q1. To what extent is the choice of a specific closed-source stack, centered on GPT-Image-1, justified beyond empirical convenience? Could the same pipeline be instantiated with ensembles including Gemini 2.5 Flash or with open-source editors at comparable quality, and what evidence supports restricting the design to closed-source components?

Q2. Given the reliance on MLLM-centric evaluation, could the study include experiments that incorporate alternative task-suitable metrics from MIG-Bench [5] and CMMD [6], as well as a VLM-based image–text similarity assessment such as CLIP-style scoring? In addition, would you consider evaluating under established protocols used by [1] and the InstructPix2Pix benchmark suite to reduce metric bias and to validate conclusions across multiple evaluation regimes?

Q3. Beyond aggregate performance, could the analysis include evidence of instruction grounding that demonstrates the fine-tuned editor actually references prompt components during editing? For example, keyword-level cross-attention or attribution maps, token-drop sensitivity curves, counterfactual prompt tests with synonym replacement, and region-wise edit-versus-background preservation metrics would make the claim concrete.

Q4. To complement the current analysis related to text encoders, would you extend comparisons beyond Qwen2.5-7B-VL to include additional MLLMs [4] and language-only embeddings, such as larger Qwen2.5 variants, with both frozen and fine-tuned settings under matched compute so that the conclusions in Table 5 are robust across architectures and scales?

[5] MIGC: Multi-Instance Generation Controller for Text-to-Image Synthesis, CVPR 2024.

[6] Rethinking FID: Towards a Better Evaluation Metric for Image Generation, CVPR 2024.

---

> ### Author Response · Authors · 2025-11-24
>
> We thank the reviewer for recognizing that:
>
> * **Clarity of Method:** Our end-to-end data curation and training workflow is presented with high clarity, particularly through the schematic that makes the methodology accessible to non-experts.
> * **Valuable Ablations:** Our systematic comparison of conditioning strategies (channel-wise vs. token-wise) and text encoders (frozen T5 vs. Qwen2.5-VL) provides useful empirical evidence on GEdit-EN and ImgEdit benchmarks.
> * **Performance Gains:** Our fine-tuning approach on the curated triplets yields concrete performance improvements, specifically achieving a score of 7.66 on GEdit-EN-full and surpassing the GPT-Image-1 baseline.
>
> We address your concerns below.
>
> ## 1. Overall clarification about the interpretation of empirical results
>
> We would like to start by thanking the reviewer for noticing the 0.17 (7.66 vs. 7.49) performance gap between our model and GPT-Image-1.
>
> However, we would like to clarify that this comparison is between our fine-tuned FluxKontext and GPT-Image-1, two models that differ significantly in architecture, training data, and other factors. To fairly assess the contribution of our GPT-IMAGE-EDIT-1.5M dataset, the more appropriate comparison is between vanilla FluxKontext and our fine-tuned FluxKontext.
>
> As presented in the paper:
>
> * FluxKontext goes from 6.26 to 7.12 on GEdit-EN, from 3.52 to 3.85 on ImgEdit (Tables 1–2, 4), and from 8.49 to 8.97 on Complex-Edit overall, with +1.06 IP over GPT-Image-1 (Table 3).
>
>
> Moreover, we additionally evaluated our model on multiple recently released reasoning- or knowledge-related image editing benchmarks:
>
> * On RISEBench, our GPT-IMAGE-EDIT-1.5M dataset improves FluxKontext from 5.8 → 12.2 (19.7 with thinking-rewrites).
> * On KRISBench, it improves FluxKontext from 49.5 → 69.0 (70.9 with thinking-style rewrites) (App. G, Tables 10–11).
>
> We hope these results help clarify the significance and impact of the GPT-IMAGE-EDIT-1.5M dataset.
>
> ## 2. “Minimal originality” and lack of filtering design
>
> First, we would like to clarify that our work is intentionally data-centric: we keep the diffusion backbones standard and instead focus on constructing a high-quality dataset distilled from a strong proprietary teacher model, along with demonstrating the empirical benefits it provides. We believe this framing aligns well with ICLR’s *Datasets and Benchmarks* track.
>
> Regarding the concern about the absence of a more sophisticated filtering mechanism for improving data quality, our goal is to maintain a minimalistic and efficient data pipeline. That said, we do incorporate meaningful safeguards:
> * we discard approximately 10% of samples through a geometric-based filter;
> * we apply instruction rewriting to ensure semantic alignment between input and output images (see our response to Reviewer vA29, “Filtering and quality control of GPT-Image-1 generations”, for details).
>
>
>
> Given the strong image-editing capabilities of GPT-Image-1, these measures are sufficient. As shown in Table 15, our dataset produces substantial improvements over raw datasets for both SD3 and Flux, with further gains achieved through instruction rewriting. In addition, we explore multiple image- and text-conditioning strategies to further enhance model performance.
>
> We believe this design delivers a clear message: distilling data from a strong proprietary model can yield significant practical benefits, and the performance gap between today’s proprietary systems and open models is likely driven largely by data quality and scale. By releasing this large-scale dataset to the community, we aim to facilitate a broad range of follow-up research — for example, improved data filtering, more advanced teacher-consensus strategies, and other methods for enhancing dataset quality.

---

> > ### Author Response · Authors · 2025-11-24
> >
> > ## 3. “Unfair comparison” and unclear closed-source versions
> >
> > For all benchmarks, we follow the official papers/leaderboards: all third-party numbers are taken directly from them; only our models and FluxKontext-dev are evaluated by us with the official scripts of each benchmark. We discuss the evaluation settings in Sec. 4.1 and App. E.
> >
> > Beyond MLLM-based benchmark scores, we now report CLIP/DINO identity and CLIP image–text similarity (see revised Table 6 for more), as the suggested MIG-Bench and CMMD are mainly designed for T2I. The key numbers are:
> >
> > | Benchmark       | Model              |  IP  |  IF  |  PQ  | Overall | CLIP-I | DINO  | CLIP-Out |
> > |-----------------|--------------------|------|------|------|---------|--------|-------|----------|
> > | GEditEN-Full    | FluxKontext        | 9.38 | 7.77 | 8.19 |  8.45   | 0.929  | 0.861 | 0.275    |
> > | GEditEN-Full    | Ours (FluxKontext) | 8.97 | 8.28 | 8.41 |  8.56   | 0.911  | 0.809 | 0.278    |
> > | ImgEdit-Full    | FluxKontext        | 9.14 | 7.79 | 8.00 |  8.31   | 0.919  | 0.746 | 0.273    |
> > | ImgEdit-Full    | Ours (FluxKontext) | 8.99 | 8.52 | 8.48 |  8.66   | 0.904  | 0.683 | 0.281    |
> > | ComplexEdit-C8  | FluxKontext        | 8.39 | 8.56 | 8.51 |  8.49   | 0.842  | 0.590 | 0.189    |
> > | ComplexEdit-C8  | Ours (FluxKontext) | 8.57 | 9.20 | 9.14 |  8.97   | 0.858  | 0.642 | 0.193    |
> >
> > These additional metrics confirm that the trends under official judges are not an artifact of a single evaluator. We emphasize that MIG-Bench and CMMD are mainly designed for T2I; where feasible, we reuse their style of embedding-based checks but keep the official editing metrics as the primary ranking signal.
> >
> > We also report parameter counts and models’ open/closed-source status in Tables 1–3 and Tables 10–12. Some suggested baselines, such as Lumina-Image 2.0, are pure T2I models without image editing functionality, therefore not compatible with our paper’s focus on instruction-guided editing.
> >
> > For Seedream-3.0, although we currently do not have access to the editing API and thus are unable to perform any evaluation, we do compare to the stronger Seedream-4.0 release on RISEBench, where our model still outperforms it, as shown in App. G, Table 10.
> >
> > ## 4. “Incremental” text-conditioning study and evidence of instruction grounding
> >
> > In the revision, we expand this part: besides T5 and Qwen2.5-VL-7B, we add Qwen2.5-VL-3B, MetaQuery connectors, and Qwen-VL+T5 hybrids (see Tables 5, 14, 18–19).
> >
> > | Text Encoder                   | GEdit-EN SC | GEdit-EN PQ | GEdit-EN Overall | ImgEdit Overall |
> > |--------------------------------|-------------|-------------|------------------|-----------------|
> > | Baseline                       | 6.98        | 7.20        | 6.26             | 3.52            |
> > | Qwen2.5-3B-VL-Instruct         | 5.98        | 7.46        | 5.84             | 3.60            |
> > | Qwen2.5-7B-VL-Instruct Metaq.  | 7.28        | 7.69        | 7.05             | 3.64            |
> > | Qwen2.5-7B-VL-Instruct         | 6.08        | 7.84        | 5.89             | 3.60            |
> > | Qwen2.5-7B-VL-Instruct+T5      | 7.91        | 7.52        | 7.24             | 3.80            |
> > | T5                             | 7.63        | 7.69        | 7.12             | 3.85            |
> > | Baseline†                      | 7.01        | 7.15        | 6.28             | 3.64            |
> > | Qwen2.5-3B-VL-Instruct†        | 7.22        | 7.41        | 6.79             | 3.66            |
> > | Qwen2.5-7B-VL-Instruct Metaq.† | 7.40        | 7.58        | 7.06             | 3.69            |
> > | Qwen2.5-7B-VL-Instruct†        | 7.25        | 7.81        | 6.87             | 3.61            |
> > | Qwen2.5-7B-VL-Instruct+T5†     | 8.08        | 7.57        | 7.45             | 3.89            |
> > | T5†                            | 8.23        | 7.75        | 7.66             | 3.90            |
> >
> > Across all variants under a frozen-encoder budget, T5 or Qwen-VL+T5 is consistently more robust than Qwen-VL alone, especially on text-heavy categories.

---

> > > ### Author Response · Authors · 2025-11-24
> > >
> > > We also add an instruction-perturbation study (Table 7) per your advice. Dropping critical spans or flipping to antonyms causes large drops, while removing non-critical spans or using synonyms has only minor impact, indicating genuine grounding in key instruction tokens.
> > >
> > > **Qwen2.5-VL-7B-Instruct**
> > >
> > > | Model                  | Add  | Adjust | Extract | Replace | Remove | Background | Style | Hybrid | Action | Overall |
> > > |------------------------|------|--------|---------|---------|--------|------------|-------|--------|--------|---------|
> > > | Qwen2.5-VL-7B-Instruct | 3.93 | 3.70   | 2.20    | 3.94    | 3.98   | 3.93       | 4.78  | 3.16   | 3.74   | 3.71    |
> > > | Drop Critical          | 3.60 | 3.30   | 2.12    | 3.09    | 1.25   | 2.55       | 4.32  | 2.48   | 3.46   | 2.91    |
> > > | Drop Noncritical       | 3.82 | 3.57   | 2.25    | 3.84    | 3.95   | 3.49       | 4.65  | 3.20   | 4.15   | 3.66    |
> > > | Drop Random            | 3.90 | 3.58   | 2.25    | 3.99    | 4.06   | 3.71       | 4.66  | 3.15   | 4.01   | 3.70    |
> > > | Synonym CF             | 3.79 | 3.70   | 2.14    | 3.90    | 3.97   | 3.58       | 4.58  | 3.17   | 3.98   | 3.65    |
> > > | Antonym CF             | 2.62 | 3.17   | 1.90    | 3.14    | 1.05   | 2.33       | 3.37  | 1.97   | 3.34   | 2.54    |
> > >
> > > **Qwen2.5-VL-7B-Instruct+T5**
> > >
> > > | Model                     | Add  | Adjust | Extract | Replace | Remove | Background | Style | Hybrid | Action | Overall |
> > > |---------------------------|------|--------|---------|---------|--------|------------|-------|--------|--------|---------|
> > > | Qwen2.5-VL-7B-Instruct+T5 | 4.01 | 3.80   | 2.72    | 4.26    | 4.13   | 3.88       | 4.78  | 3.33   | 4.11   | 3.89    |
> > > | Drop Critical             | 3.81 | 3.29   | 2.67    | 3.53    | 1.43   | 2.91       | 4.53  | 2.45   | 3.08   | 3.08    |
> > > | Drop Noncritical          | 3.95 | 3.50   | 2.89    | 4.14    | 3.96   | 3.84       | 4.77  | 3.33   | 4.48   | 3.87    |
> > > | Drop Random               | 3.88 | 3.59   | 2.99    | 4.16    | 3.99   | 3.87       | 4.87  | 3.38   | 4.41   | 3.90    |
> > > | Synonym CF                | 3.91 | 3.51   | 2.45    | 4.19    | 4.05   | 3.84       | 4.80  | 3.51   | 4.37   | 3.85    |
> > > | Antonym CF                | 2.92 | 3.20   | 1.99    | 2.82    | 1.10   | 2.28       | 3.42  | 2.29   | 3.27   | 2.59    |
> > >
> > > **T5**
> > >
> > > | Model           | Add  | Adjust | Extract | Replace | Remove | Background | Style | Hybrid | Action | Overall |
> > > |-----------------|------|--------|---------|---------|--------|------------|-------|--------|--------|---------|
> > > | T5              | 4.05 | 3.72   | 3.18    | 4.27    | 3.99   | 3.95       | 4.85  | 3.59   | 4.54   | 3.91    |
> > > | Drop Critical   | 3.94 | 3.43   | 2.91    | 3.61    | 1.59   | 2.83       | 4.66  | 2.68   | 3.49   | 3.14    |
> > > | Drop Noncritical| 4.03 | 3.78   | 3.03    | 4.26    | 3.86   | 3.90       | 4.79  | 3.35   | 4.59   | 3.86    |
> > > | Drop Random     | 4.11 | 3.84   | 3.10    | 4.28    | 4.03   | 3.91       | 4.80  | 3.30   | 4.59   | 3.92    |
> > > | Synonym CF      | 4.09 | 3.83   | 2.96    | 4.27    | 4.00   | 3.87       | 4.80  | 3.39   | 4.51   | 3.87    |
> > > | Antonym CF      | 2.67 | 3.07   | 1.94    | 2.59    | 1.02   | 2.32       | 3.51  | 2.42   | 3.31   | 2.42    |
> > >
> > > Performance drops sharply when critical spans are removed or flipped to antonyms, but stays relatively stable under non-critical drops or synonyms. This, together with our ablation study about different text-conditioning methods, shows that the editor actually uses the semantic content of the instructions.
> > >
> > > Jointly fine-tuning 7B–72B VLMs with a 12B diffuser is beyond our current compute/data budget and the scope of this work, so we explicitly frame deep fusion and larger VLMs as future directions.

---

> ### Comment · Reviewer_vipp · 2025-11-26
>
> **1. Clarification**
>
> I appreciate the authors' clarification regarding the performance comparison; however, fine-tuning a smaller "student" model using high-quality synthetic data distilled from a vastly superior "teacher" model (GPT-Image-1) is a well-established strategy that guarantees performance gains, mirroring the trajectory of early open-source LLMs [A, B] which bridged the gap with proprietary models through similar distillation. If this approach was indeed inspired by such strategies to narrow the divide between open and proprietary models, I strongly suggest explicitly stating this in the paper to prevent any misunderstanding regarding the novelty of the methodology. Consequently, I view the reported improvements as a straightforward adaptation of this known recipe to the image editing domain; while this confirms that bridging the data scale and quality gap improves performance, it represents a reproduction of the proprietary model’s capabilities via brute-force data scaling rather than a distinct methodological innovation or technical contribution to the field.
>
> **2. Concerns Regarding Originality, Licensing, and Reproducibility**
>
> Regarding the dataset's contribution, while I agree that GPT-IMAGE-EDIT-1.5M could facilitate open research, critical uncertainties about its practical availability remain unaddressed. Despite the details in Appendix B, it is ambiguous whether distributing 1.5 million images generated by GPT-Image-1 to train competitive models fully complies with OpenAI's Terms of Use or safety policies regarding NSFW content; if legal or ethical constraints force a partial or restricted release, the primary contribution of this work would be severely compromised. Furthermore, my review of the supplementary material revealed only skeleton code without any actual data subsets or fine-tuned checkpoints, which makes it difficult to trust that the dataset will be fully and transparently released. I request explicit confirmation on whether the entire dataset can be legally distributed to ensure true reproducibility.
>
> **3. Unfair/Unclear Comparison**
>
> Thank you for clarifying and sharing the results. The additional metrics and explanations regarding baseline selection have fully addressed my concerns on this matter.
>
> **4. Incremental Text-Conditioning Study**
>
> While I acknowledge the additional experiments with Qwen2.5-VL-3B and hybrid combinations, my concerns regarding the narrow scope and mechanism-level insight remain unresolved.
>
> * **Lack of Architectural Diversity:** The added experiments are restricted to marginal variations within the specific Qwen2.5-VL family. To demonstrate that the findings are not specific to Qwen2.5, it is necessary to evaluate distinct SOTA MLLM architectures (e.g., FastVLM, InternVL) as originally requested to verify consistent improvements across representative models.
>
> * **Pretraining Bias:** The sustained superiority of T5 is unsurprising given that the backbone models (SD3/FLUX) utilized T5 during their pretraining, creating a strong initialization bias. To rigorously verify the dataset's value independent of this architectural coupling, I request an experiment extending to Qwen-Image (or a similar MMDiT SOTA) which natively relies solely on an MLLM text encoder. This would clarify whether the dataset contributes to performance improvements in MLLM-native architectures without the confounding factor of T5 pretraining.
>
> If providing other results related to **[W4. Limitations]** is feasible, I would appreciate seeing the results.
>
> While the clarification on baselines is appreciated, my concerns regarding the distillation's triviality, limited text conditioning scope, and dataset licensing/release remain. I am open to raising my score if the authors provide concrete evidence ensuring full reproducibility and broader experimental validation.
>
> **References:**
>
> [A] Vicuna: An open-source chatbot impressing gpt-4 with 90%* chatgpt quality
>
> [B] Alpaca: A Strong, Replicable Instruction-Following Model

---

> > ### Author Response · Authors · 2025-12-03
> >
> > We thank Reviewer *vipp* again for the careful follow‑up and for engaging seriously with our work. In this comment we address the remaining concerns on **(1) originality** and **(2) licensing and reproducibility**. We respond on the text‑encoder study in a separate comment.
> >
> > **1. Originality and the role of distillation in image editing**
> >
> > We agree that teacher–student distillation is now standard in the LLM literature (e.g., Vicuna, Alpaca). Our goal is not to claim novelty for distillation itself, but to establish a concrete, previously missing result in **instruction‑guided image editing** and to position it clearly in ICLR’s **Datasets and Benchmarks** track.
> >
> > **Transition from complex expert pipelines to a simple general‑purpose pipeline.**
> > Prior synthetic editing datasets (e.g., HQ‑Edit) also rely on synthetic images but typically adopt heavy, multi‑stage expert pipelines and task‑specific filters to make data usable for training strong editors. In contrast, we show that a single strong general‑purpose editor (GPT‑Image‑1), together with a minimal, model‑agnostic curation pipeline (pad‑and‑crop, simple geometric filtering, and instruction rewriting), is already sufficient to produce, to the best of our knowledge, a million‑scale, fully synthetic, unified editing dataset that:
> >
> > * unifies OmniEdit, HQ‑Edit, and UltraEdit into one schema;
> > * achieves high instruction–image alignment using only simple, generic filters rather than task‑specific expert pipelines such as OmniEdit.
> >
> > This demonstrates that, in the editing regime, earlier ad‑hoc expert pipelines can largely be replaced by a much simpler general‑purpose recipe without sacrificing performance.
> >
> > **Substantial leap in open‑source performance and bridging to closed models.**
> > Instruction‑guided editing is inherently more difficult than text‑to‑image: the model must follow instructions and maintain perceptual quality while preserving global layout and identity. Our work presents a fully open‑source approach that is among the strongest open‑source editors and can even match or surpass GPT‑Image‑1 across several benchmarks, including:
> >
> > * **GEdit‑EN** and **ImgEdit**, traditional task‑oriented benchmarks;
> > * **Complex‑Edit**, with extremely long, compositional instructions intended as a stress test for editing;
> > * **RISEBench** and **KRISBench**, two recent benchmarks designed to evaluate reasoning‑ and knowledge‑related image editing.
> >
> > This is not just “brute‑force scaling”: it is direct evidence that a simple general‑purpose synthetic editing dataset can materially upgrade a strong open editor in the editing setting, without any architectural changes, and substantially narrow the gap to GPT‑Image‑1.
> >
> > **Positioning within the Datasets & Benchmarks track.**
> > Our submission is explicitly to the **Datasets and Benchmarks** track. The core contribution is the dataset and the insights it enables. Its scale, unified design, and practical usefulness, together with systematic experiments on conditioning, text encoders, and instruction perturbations,
> >
> > * show that high‑quality synthetic editing triplets significantly improve a strong open editor and reduce the gap to proprietary models;
> > * indicate that many earlier hand‑crafted expert pipelines for synthetic editing are no longer necessary once a strong general‑purpose editor is available.
> >
> > We will make this positioning even more explicit in the final version so that the novelty is clearly framed at the level of the image‑editing dataset and its demonstrated impact.
> >
> > **2. Licensing, dataset release, and reproducibility**
> >
> > We have carefully checked OpenAI’s Terms of Use and aligned our release plan accordingly. As stated in our previous rebuttal and in the manuscript, we intend to release GPT‑IMAGE‑EDIT‑1.5M under a non‑commercial license that is consistent with the restrictions imposed by OpenAI and all other data sources we used.
> >
> > **Rights and constraints.**
> > Under the current Terms, users retain rights over their inputs and are assigned rights over generated outputs, provided they continue to comply with the Terms. A key constraint is that outputs from GPT‑Image‑1 must not be used to train competing commercial models. Any downstream user of GPT‑IMAGE‑EDIT‑1.5M must also comply with these Terms.
> >
> > **Our release policy.**
> > To reflect this, we will release GPT‑IMAGE‑EDIT‑1.5M under **CC BY‑NC 4.0**, explicitly limiting it to non‑commercial research and evaluation. In the dataset card and Ethics section, we clearly remind users that:
> >
> > * they must obey both CC BY‑NC 4.0 and OpenAI’s Terms when using GPT‑Image‑1‑derived content;
> > * the dataset is not intended for training competing commercial services.

---

> > > ### Author Response · Authors · 2025-12-03
> > >
> > > **2. Licensing, dataset release, and reproducibility (continued)**
> > >
> > > **Practical availability and ICLR limits.**
> > > ICLR 2026 restricts the supplementary ZIP file to 100 MB, so we cannot include our full 1.5M-scale dataset or the >10 GB checkpoints. The current supplementary material therefore includes:
> > >
> > > * a complete, runnable codebase and data schema;
> > > * a small visual subset illustrating the data format and curation effects.
> > >
> > > **Timing and scope of the full release.**
> > > To preserve double-blind review, we cannot publicly link the full dataset and checkpoints during the review phase. However, we have already prepared a complete release package (data + checkpoints) under **CC BY-NC 4.0**, and we will make it fully available as soon as anonymity is no longer required (camera-ready or earlier, in line with program-chair guidance).
> > >
> > > Given this setup and to the best of our knowledge, we do not see any clause in the current OpenAI Terms that forbids non-commercial research redistribution of GPT-Image-1-derived images under a license that (i) is non-commercial and (ii) requires downstream users to comply with those Terms. If there is a specific provision we have missed or misunderstood, we are happy to adjust the release plan or clarify the legal language more explicitly in the paper.
> > >
> > > **3. Text encoders: new MLLM baselines and narrowed conclusions**
> > >
> > > We agree that our original wording comparing “T5 vs. MLLM embeddings” could sound too broad. Following the reviewer’s suggestions, we (i) added more diverse MLLM-family encoders, and (ii) stated the scope of our conclusions more precisely.
> > >
> > > **New experiments with diverse MLLM encoders.**
> > > Beyond Qwen2.5-VL, we now replace the T5 encoder with several distinct MLLM encoders under the same shallow connector and training budget, keeping the FluxKontext backbone fixed. The main results are summarized below:
> > >
> > > | Text Encoder              | GEdit-EN SC | GEdit-EN PQ | GEdit-EN Overall | ImgEdit Overall |
> > > | ------------------------- | ----------- | ----------- | ---------------- | --------------- |
> > > | Baseline  | 6.98        | 7.20        | 6.26             | 3.52            |
> > > | Qwen2.5-3B-VL             | 5.98        | 7.46        | 5.84             | 3.60            |
> > > | Qwen2.5-7B-VL             | 6.08        | 7.84        | 5.89             | 3.60            |
> > > | FastVLM-7B                | 5.84        | 7.67        | 5.86             | 3.56            |
> > > | InternVL-3.5-8B           | 6.24        | 7.81        | 6.28             | 3.64            |
> > > | Qwen3-8B-VL               | 6.32        | 7.79        | 6.31             | 3.68            |
> > > | T5                        | 7.63        | 7.69        | 7.12             | 3.85            |
> > >
> > > Within this matched-compute, shallow-fusion setting, none of the tested MLLM encoders surpasses T5. The strongest MLLM (Qwen3-8B-VL) roughly matches the baseline but still lags behind T5, especially on ImgEdit.
> > >
> > > **More precise statement of our conclusion.**
> > > Based on these results, we will state our claim in the paper as follows:
> > >
> > > > For MMDiT-style editors that were pre-trained with T5, simply replacing the native T5 encoder with frozen MLLM embeddings via a shallow linear connector does not yield performance gains; in this setting, retaining the T5 encoder is a more robust choice.
> > >
> > > We also explicitly note that this is strongly influenced by pre-training: SD3/Flux-family models were originally pre-trained with T5, so any alternative encoder starts from a disadvantage unless one invests in substantial re-training or deep cross-modal fusion.
> > >
> > > **Limitations for MLLM-native architectures (e.g., Qwen-Image).**
> > > We fully agree that our conclusion does **not** extend to architectures that are natively MLLM-centric, such as Qwen-Image-style MMDiTs. These MLLM-native architectures (e.g., Qwen-Image)
> > >
> > > * follow very different pre-training recipes,
> > > * operate at much larger parameter scales (around 27B parameters),
> > >
> > > making them fundamentally incomparable to the T5-pretrained MMDiT editors evaluated in this work.
> > >
> > > Under our current compute budget (8×A100-80G), fully fine-tuning such stacks at 1024² resolution is not feasible. We will state this limitation clearly and explicitly list “MLLM-native architectures and deep cross-modal fusion for editing” as important future work.
> > >
> > > Overall, our text-encoder study is meant to provide **practical guidance for T5-pretrained MMDiT editors in the shallow-fusion regime**, rather than to claim a universal statement about all MLLM-based encoders. Within a realistic compute and anonymity budget, we believe we have addressed all concerns we can directly act on (fairness, metrics, grounding, and licensing). The only remaining item we did not provide, full fine-tuning of a Qwen-Image–scale stack, is beyond our current feasibility and, in our view, not a requirement for this Datasets & Benchmarks submission, but rather a natural direction for future work building on GPT-IMAGE-EDIT-1.5M.

---

### Official Review · Reviewer_vA29 · 2025-10-31

**Soundness:** 3
**Presentation:** 3
**Contribution:** 3
**Rating:** 6
**Confidence:** 3

**Summary:**

This paper aims to bridge the gap between closed-source multimodal image editing models (e.g., GPT-Image-1) and open-source research by introducing GPT-IMAGE-EDIT-1.5M—a publicly available dataset comprising over 1.5 million high-quality image editing triplets (original image, editing instruction, target image). The dataset integrates data from OmniEdit, HQEdit, and UltraEdit, and further enhances instruction following and perceptual quality through output regeneration and instruction rewriting, while deliberately preserving the challenges associated with identity preservation.

**Strengths:**

1. The paper presents the first publicly released, million-scale image editing dataset with unified formatting and high alignment between instructions and outputs, significantly advancing research in image editing.
2. The paper provides an in-depth investigation of channel-wise vs. token-wise conditioning, clearly demonstrating the superiority of token-wise conditioning in context-aware editing tasks.
3. Models trained on the proposed dataset achieve state-of-the-art (SOTA) performance across multiple standard benchmarks, validating the effectiveness of the approach.
4. The paper is clearly written and well-structured.

**Weaknesses:**

1. After using GPT-Image-1 to generate edited images, did the authors apply any filtering or quality screening to these newly generated images? For instance, were samples with generation failures, severe distortions, or complete misalignment with the instruction removed? If filtering was performed, please detail the criteria and procedure used.
2. Why was knowledge distillation conducted exclusively from GPT-Image-1, rather than aggregating outputs from multiple top-tier closed-source or open-source models to construct a more diverse and robust “consensus” dataset? For example, one could employ voting or fusion strategies across multiple model outputs.
3. Are the quantitative results in the paper solely based on automatic evaluation by MLLMs? Did the authors conduct any human evaluation? Relying exclusively on MLLM-based scoring may introduce biases or hallucinations inherent to the MLLM itself.

**Questions:**

See Weaknesses

---

> ### Author Response · Authors · 2025-11-24
>
> We thank the reviewer for recognizing that
>
> - our GPT-IMAGE-EDIT-1.5M process relies not only on large data scale but also on high data quality in terms of unified formatting and strong alignment, which significantly advances research in image editing;
> - our in-depth investigation of different image conditioning methods shows the superiority of token-wise conditioning in context-aware editing tasks; and
> - the model fine-tuned on our GPT-IMAGE-EDIT-1.5M achieves SOTA performance across multiple benchmarks, further validating the effectiveness of our dataset.
>
> Below we address the three weaknesses point by point.
>
> ## 1. Filtering and quality control of GPT-Image-1 generations
>
> We apologize for not clearly describing this detail in the original submission. Our curation includes a geometric-based filter followed by instruction rewriting.
>
> ### Geometric-based filtering
>
> As GPT-Image-1 only supports three aspect ratios (1:1, 3:2, 2:3) while OmniEdit / HQEdit / UltraEdit have arbitrary ratios, we apply a pad–generate–crop pipeline: we pad each input to the nearest supported ratio, edit with GPT-Image-1, then crop back to the original field of view and resolution.
>
> We automatically discard outputs with residual blank borders after cropping using a simple rule-based detector, which removes ~10% of GPT-Image-1 generations.
>
> As shown in App. B.1/B.6, Fig. 5, with geometric-based filtering we ensure that the overall layout of the input and output images is aligned spatially. We then further align the input and output images at a more fine-grained semantic level.
>
> ### Instruction rewriting
>
> For the filtered samples, we further enhance them by employing instruction rewriting to align the actual changes in the output images and the editing instructions on a semantic level.
>
> In App. D.2, we discuss the process of regenerating editing instructions with a VLM (GPT-4o) based on the input images and the re-edited output images, which is an approach well adopted in recent editing datasets such as **“HQ-Edit: A High-Quality Dataset for Instruction-Based Image Editing”** [1].
>
> ## 2. Why only GPT-Image-1 as the image editor, instead of a multi-model consensus in the data pipeline?
>
> We would like to clarify that our pipeline is model-agnostic. For preparing this specific paper, the selection of GPT-Image-1 is mainly driven by quality and simplicity:
>
> - At the time of data collection, GPT-Image-1 was clearly ahead of widely available editing models, as reflected by performance rankings on GEdit-EN / ImgEdit, providing a strong, stable reference distribution.
> - Using one image editor keeps the pipeline simple and more aligned with our motivation: to swap in another editing model, one only needs to change a single component in Fig. 2, without redesigning the rest.
>
> We also agree that multi-model consensus is very promising, and we plan to explore model ensembles (e.g., GPT-Image-1 + strong open editors, or even with the latest Nano Banana Pro) with VLM-based voting in follow-up work.
>
> **References**
>
> [1] *HQ-Edit: A High-Quality Dataset for Instruction-Based Image Editing*.

---

> > ### Author Response · Authors · 2025-11-24
> >
> > ## 3. Reliance on MLLM-based evaluation and lack of human study
> >
> > Firstly, we would like to clarify that we follow the official evaluation protocols of all benchmarks we use — GEdit-EN, ImgEdit, Complex-Edit, RISEBench, and KRISBench — which are all based on VLM/MLLM judges by design. We agree that this may lead to bias, so we additionally collect the following metrics.
> >
> > ### Non-VLM based evaluation
> >
> > We add CLIP image–image similarity, DINO features, and CLIP image–text similarity as complementary metrics (Table 6), and observe that improvements under MLLM scores are mirrored by improvements under these independent, vision-centric metrics:
> >
> > | Benchmark       | Model              |   IP |   IF |   PQ | Overall | CLIP-I | DINO  | CLIP-Out |
> > |-----------------|--------------------|------|------|------|---------|--------|-------|----------|
> > | GEditEN-Full    | FluxKontext        | 9.38 | 7.77 | 8.19 |   8.45  | 0.929  | 0.861 | 0.275    |
> > | GEditEN-Full    | Ours (FluxKontext) | 8.97 | 8.28 | 8.41 |   8.56  | 0.911  | 0.809 | 0.278    |
> > |                 |                    |      |      |      |         |        |       |          |
> > | ImgEdit-Full    | FluxKontext        | 9.14 | 7.79 | 8.00 |   8.31  | 0.919  | 0.746 | 0.273    |
> > | ImgEdit-Full    | Ours (FluxKontext) | 8.99 | 8.52 | 8.48 |   8.66  | 0.904  | 0.683 | 0.281    |
> > |                 |                    |      |      |      |         |        |       |          |
> > | ComplexEdit-C8  | FluxKontext        | 8.39 | 8.56 | 8.51 |   8.49  | 0.842  | 0.590 | 0.189    |
> > | ComplexEdit-C8  | Ours (FluxKontext) | 8.57 | 9.20 | 9.14 |   8.97  | 0.858  | 0.642 | 0.193    |
> >
> > These additional metrics confirm that the trends under official judges are not an artifact of a single evaluator.
> >
> > ### Human experiments
> >
> > We are actively working on a small-scale human study for the revised version. However, given the overhead of building evaluation tools and obtaining IRB approval for these human-related experiments, we are uncertain whether this can be fully finalized before the rebuttal deadline. If not, we will make sure such results are presented in the final version.
> >
> > Furthermore, to clearly mark the potential bias introduced by MLLM-based judges, we explicitly list **“reliance on MLLM-based scoring”** as a limitation in Sec. 5.

---

> > > ### Comment · Reviewer_vA29 · 2025-11-24
> > >
> > > Thank you to the authors for their responses, which have fully addressed my concerns. I vote to accept this paper.

---

### Official Review · Reviewer_aoHw · 2025-10-31

**Soundness:** 3
**Presentation:** 3
**Contribution:** 3
**Rating:** 8
**Confidence:** 4

**Summary:**

This paper addresses the performance gap between proprietary instruction-guided image editors (like GPT-Image-1) and their open-source counterparts. The authors identify the primary bottleneck as the lack of large-scale, high-quality training data, as existing datasets often suffer from poor alignment or overly simplistic instructions.

The main contribution is GPT-IMAGE-EDIT-1.5M, a new, publicly available dataset of over 1.5 million editing triplets. This dataset is curated by systematically unifying and refining three existing datasets (OmniEdit, HQEdit, UltraEdit). The refinement pipeline uniquely leverages proprietary models—using GPT-Image-1 for output regeneration and GPT-4O/5 for instruction rewriting—to significantly enhance instruction-following (IF) and perceptual quality (PQ). Crucially, the dataset intentionally preserves challenging identity preservation (IP) cases, reflecting realistic complexities

**Strengths:**

1. The primary strength is the dataset itself. It is large (1.5M samples)  and meticulously curated. The pipeline, which uses GPT-Image-1 for regeneration and GPT-4O for rewriting, directly addresses the known weaknesses (poor alignment, simple instructions) of prior datasets. More important, The authors commit to releasing the dataset, models, and code, which is a significant service to the community.
2. The decision to intentionally preserve difficult identity preservation (IP) cases is a significant and important one. This strategic choice makes the dataset more realistic.
3. The paper provides a clear and impactful conclusion: token-wise conditioning is superior to channel-wise conditioning for complex editing. The ablation in Table 4 is a powerful piece of evidence, showing that channel-wise models regress on this challenging data while token-wise models improve. This is a valuable directive for the community.
4. The counter-intuitive finding that a frozen T5 text-only encoder is a more robust choice than a frozen MLLM (Qwen2.5-VL) is a very practical and useful insight for open-source development. The analysis correctly identifies that shallow fusion is a bottleneck, pointing toward deep fusion as a necessary area for future work.

**Weaknesses:**

1. The paper's contribution is primarily empirical and data-centric. It does not propose a new model architecture, conditioning mechanism, or fusion strategy. Instead, it offers a thorough benchmark of existing components (SD3-IP2P, Flux, T5, Qwen-VL). While this analysis is valuable, the paper is an "analysis of what works" rather than a "proposal of a new method."

**Questions:**

None

---

> ### Author Response · Authors · 2025-11-24
>
> We sincerely thank the reviewer for the careful and positive assessment of our work, and for recognizing that
>
> - our attempt to bridge the performance gap between proprietary image editors and open‑source counterparts by uniquely leveraging proprietary models for image regeneration and instruction rewriting significantly improves instruction‑following (IF) and perceptual quality (PQ);
> - as shown in our experiments, our dataset helps to significantly mitigate the known weaknesses (e.g., poor alignment, simple instructions) of prior datasets;
> - beyond the dataset itself, our analysis of different approaches to image conditioning and text conditioning is practical and insightful, with strong support from empirical evidence.
>
> Regarding the major contribution of this paper, we would like to clarify that our work is submitted to ICLR’s *Datasets and Benchmarks* Topic. More specifically, by offering the research community this large‑scale, useful, and accessible dataset, we aim to show that the gap between proprietary systems and open‑source models can be effectively bridged primarily by improving data, together with systematic choices of conditioning and text encoders.
>
> To summarize, our main contributions are:
>
> - **Unified, simple data pipeline.** We build GPT‑IMAGE‑EDIT‑1.5M by leveraging a single strong general‑purpose editor with a minimal set of simple filters and no ad hoc designs for each editing task, effectively replacing the heterogeneous “expert” pipelines previously designed for different editing tasks.
>
> - **Strong editors trained with 1.5M samples.** With our dataset of 1.5M samples, fine‑tuning standard FluxKontext editors on our dataset achieves open‑source SOTA or near‑SOTA on GEdit‑EN, ImgEdit, and Complex‑Edit, and open‑source SOTA on RISEBench and KRISBench. This performance is further boosted by thinking‑style rewriting as a test‑time scaling method, which even exceeds GPT‑Image‑1, the proprietary model we mainly used to collect training data.
>
> - **Actionable design guidance.** We carry out systematic comparisons across conditioning paradigms (channel‑wise vs. token‑wise) and text encoders (T5 vs. MLLM with shallow fusion), and distill concrete training recommendations for practitioners building instruction‑guided editors on top of our dataset.
>
> Based on the feedback from reviewers, we have also revised the following parts of our manuscript:
>
> - providing more details of the regeneration and pad‑and‑crop pipeline, including quantitative statistics and visual examples (App. B);
> - adding CLIP/DINO‑based identity and image–text alignment metrics (Table 6), showing that the trends under MLLM‑based scores are consistent with vision‑centric metrics; and
> - expanding ablations on text encoders and instruction perturbations (Tables 5, 7, 14, 18–19) to make our design recommendations more actionable.

---

> > ### Comment · Reviewer_aoHw · 2025-11-26
> >
> > I appreciate the author's efforts on rebuttal, my concerns are resolved and I have ultimately decided to maintain the current rating.

---

### Author Response · Authors · 2025-12-02

Dear Reviewers, ACs, and SACs,

We thank all reviewers for their detailed feedback. Three reviewers (*aoHw, vA29, iuXW*) support acceptance, highlighting the clarity of our data pipeline, the scale and alignment of GPT‑IMAGE‑EDIT‑1.5M, and the value of our ablations. Reviewer *vipp*, despite a lower rating, also acknowledges the strengths of our dataset construction, the comparative experiments, and the clear performance gains for open‑source editors.

**Overall agreement**

Across all four reviews, there is broad agreement on three main points:

* **First large‑scale unified editing dataset.**
  GPT‑IMAGE‑EDIT‑1.5M is, to the best of our knowledge, the first million‑scale unified dataset for instruction‑guided image editing constructed from a single general‑purpose generative model rather than the multi‑stage expert pipelines used in prior synthetic editing datasets. Reviewers note that this simple, model‑agnostic pipeline still achieves strong instruction–image alignment (*aoHw, vA29, iuXW, vipp*).

* **Strong performance among open‑source editors.**
  Fine‑tuning FluxKontext‑12B on our dataset yields consistent gains over the vanilla model across editing benchmarks. The resulting model becomes one of the strongest open‑source editors, substantially narrowing the gap to GPT‑Image‑1 on GEdit‑EN, ImgEdit, and Complex‑Edit, and reaching state‑of‑the‑art performance among open‑source models on RISEBench and KRISBench (*vA29, iuXW, vipp*).

* **Actionable guidance for practitioners.**
  Our comparisons of channel‑wise vs token‑wise conditioning and different frozen text encoders provide concrete recommendations for building instruction‑based editors on public MMDiT backbones (*aoHw, vA29, iuXW, vipp*).

**Common concerns and our responses**

During rebuttal we added clarifications and new experiments on the main concerns:

* **Originality and positioning** (*aoHw, vA29, iuXW, vipp*).
  We frame the work in the Datasets & Benchmarks track, showing that a simple GPT‑based synthetic editing pipeline can replace earlier multi‑stage expert pipelines and that distilling from a strong teacher measurably improves a strong open editor.

* **Evaluation fairness and metrics** (*vA29, iuXW, vipp*).
  We clarified evaluation settings and added CLIP image–image similarity, DINO identity features, and CLIP image–text alignment, showing that gains under MLLM judges are mirrored by vision‑centric metrics.

* **Licensing, release, and reproducibility** (*iuXW, vipp*).
  We added a licensing table and clarified that GPT‑IMAGE‑EDIT‑1.5M will be released under **CC BY‑NC 4.0**, reminding users that they must also comply with OpenAI’s Terms when using GPT‑Image‑1‑derived content. We will release data, checkpoints, and code after the review period.

* **Proprietary dependence and open‑stack variants** (*iuXW, vipp*).
  GPT‑Image‑1 and GPT‑4o/5 are used only as data collectors; the trained editor is fully open‑source. We provide all curation scripts so others can construct an open‑only variant. We also added an ablation replacing GPT‑5 with open‑weight Qwen3‑VL‑32B for inference‑time rewriting, which yields comparable or stronger results, indicating that our conclusions do not depend on a specific proprietary VLM.

* **Text encoders and instruction grounding** (*aoHw, vA29, vipp*).
  We expanded the text‑encoder study with additional MLLM encoders and Qwen‑VL+T5 hybrids. Under matched compute, none of these alternatives outperform T5 for T5‑pretrained MMDiT editors, and we narrow our claims to this regime. We also added an instruction‑perturbation study showing sharp drops when critical spans are removed or flipped, while performance is stable under non‑critical drops or synonym replacements, confirming genuine grounding on key instruction tokens.

* **Compute limits and scope** (*iuXW, vipp*).
  We summarize the training budget (about 1500 A100‑80G GPU‑hours at 1024²) and clarify that fully fine‑tuning a Qwen‑Image‑scale, MLLM‑native unified stack is beyond our current compute and anonymity budget. We therefore treat such deeply fused architectures as natural follow‑up work enabled by GPT‑IMAGE‑EDIT‑1.5M, rather than as part of this submission.

**Summary of revisions**

* Added CLIP/DINO identity and CLIP image–text alignment metrics, plus new evaluations on RISEBench and KRISBench.
* Expanded text‑encoder ablations and instruction‑perturbation analyses.
* Added an open‑weight Qwen3‑VL inference‑time rewriting ablation and a concise compute summary for FluxKontext training.
* Clarified dataset licensing (CC BY‑NC 4.0), its interaction with OpenAI’s Terms, and our post‑review plan to release the full dataset and checkpoints.
* Refined the introduction, related work, and conclusion to present the paper as a data‑centric Datasets & Benchmarks contribution and to limit claims to the regime we directly study.

We address each reviewer’s comments point by point below and welcome further discussion. Thank you!

---

### Note · Authors · 2026-02-23

I have read and agree with the venue's withdrawal policy on behalf of myself and my co-authors.

---

### Meta-Review · Area_Chair_g3nf · 2026-01-06

**Summary:**

The major concerns raised by the reviewers include:
1. The lack of substantial improvement due to data quality issues and insufficient filtering.
2. Potentially unfair comparisons between different methods.
3. Limitations related to reproducibility and data release.
4. Limited technical contribution.
In particular, whether the proposed dataset can be released is a central concern, given that the primary contribution of this work lies in the curated dataset.

**Reviewer Concerns:**

Most concerns are adequately addressed, with the exception of the limitations on data release and the limited novelty. In particular, data release remains the key decision factor for this paper given the limited technical contribution. Although the authors argue that the dataset can be released under OpenAI terms of usage, it remains unclear whether this is feasible in practice.

**Reviewer Scores:**

Given the clear limitation of this manuscript, namely, whether the dataset can be released, further discussion is unlikely to be productive unless a concrete commitment to data release can be obtained.

---

### Decision · Program_Chairs · 2026-01-26

Reject